# Carbon pricing and planetary boundaries

Gustav Engström [1,2 ✉], Johan Gars[1,2], Chandra Krishnamurthy[1,3], Daniel Spiro[4], Raphael Calel [5],
Therese Lindahl [1,6] & Badri Narayanan [7]

Human activities are threatening to push the Earth system beyond its planetary boundaries, risking catastrophic and irreversible global environmental change. Action is urgently needed, yet well-intentioned policies designed to reduce pressure on a single boundary can lead, through economic linkages, to aggravation of other pressures. In particular, the potential policy spillovers from an increase in the global carbon price onto other critical Earth system processes has received little attention to date. To this end, we explore the global environmental effects of pricing carbon, beyond its effect on carbon emissions. We find that the case for carbon pricing globally becomes even stronger in a multi-boundary world, since it can ameliorate many other planetary pressures. It does however exacerbate certain planetary pressures, largely by stimulating additional biofuel production. When carbon pricing is allied with a biofuel policy, however, it can alleviate all planetary pressures.

[1] Beijer Institute of Ecological Economics, Royal Swedish Academy of Sciences, 10405 Stockholm, Sweden. [2] GEDB, Royal Swedish Academy of Sciences, 10405 Stockholm, Sweden. [3] Swedish University of Agricultural Sciences (SLU), 90187 Umeå, Sweden. [4] Department of Economics, Uppsala University, 751 20 Uppsala, Sweden. [5] Georgetown University, Washington, DC 20057, USA. [6] Stockholm Resilience Centre, Stockholm University, 10691 Stockholm, Sweden. [7] School of Environmental and Forestry Sciences, University of Washington Seattle, Seattle, WA 98074, USA. ✉email: gustav.engstrom@beijer.kva.se

The Earth has been in a remarkably steady state over the last 10,000 years, but human activities since the industrial revolution are now starting to threaten its balance. Rockström and colleagues[1,2] have developed a list of nine Earth system processes (ESPs) that are critical to maintaining a stable global environment: biogeochemical flows, ocean acidification, freshwater use, land-use change, biodiversity loss, atmospheric aerosol loading, ozone depletion, and chemical pollution. There are also nine corresponding planetary boundaries beyond which mankind may not proceed without risking potentially catastrophic consequences[2,3]. Even as these planetary boundaries are gaining policy recognition[4,5], the complexity of the many interlocking processes can seem to present decision makers with an unnavigable obstacle course.

In this paper, we develop a stylized yet empirically grounded framework for analyzing these interlocking processes. We assess the environmental consequences of a global carbon pricing policy in a multi-boundary world[2,3]. While it seems unlikely that a global carbon price would be adopted in the near term, this policy serves as a useful proxy for more stringent climate policy in general. Consequently, our analysis may be interpreted as identifying which ESPs are of particular concern when (if) climate policy becomes more ambitious. Carbon pricing is frequently viewed as a matter of applying the brakes on greenhouse gas emissions, but a multi-boundary perspective alerts us to the possibility that it might inadvertently redirect economic activities in ways that exacerbate (or alleviate) other harmful environmental pressures. For instance, a carbon tax targeting fossil fuels will most likely stimulate the production of biofuels, such as palm oils, leading to both additional land-conversion and adverse effects on biodiversity[6,7]. Consequently, a specific policy that moves the Earth system away from one boundary could inadvertently move it toward another[8]. A better analogy to the problem of policy-making with multiple planetary boundaries may, therefore, be that of parallel parking, where the challenge is to simultaneously respect boundaries on all sides, ensuring a safe operating space for global societal development[3].

Prior literature has mainly focused on studying how policies affect a single ESP in isolation[9,10], or on developing large-scale computational models to provide counterfactual simulations for two closely related ESPs, such as land-use/deforestation, or climate change/food security[11–14]. The purpose of the present paper differs from these in at least two important aspects. First, the scope of our analysis involves all planetary boundaries and encompasses a majority of the underlying drivers. Second, our intentions are to provide a framework that is rich enough to let researchers investigate and discover the complex ways in which policies can interact with multiple ESPs, while also being simple enough to yield a qualitative understanding of the intended and unintended side effects of global environmental policies. Our objective is thus to establish a high-level understanding of how economic markets drive and interact with the planetary boundaries.

To this end, we present a new global economic policy analysis model developed specifically for the analysis of the economic drivers threatening the safe operating space for global societal development. The model includes the quantitatively most important economic drivers of environmental change, is calibrated using data from one of the most widely-used sources for economic modeling, the Global Trade Analysis Project (GTAP)[15], and links these economic sectors to the ESPs highlighted in the planetary boundaries framework. We focus exclusively on economic linkages and for the sake of transparency, thus exclude from any direct linkages between biophysical processes which has previously been assessed in e.g. ref. [16]. The model is further designed to both enable replication and independent assessment

of presented findings (attempting to address the transparency concerns often directed at large-scale IAMs[17,18]), as well as further exploration of the effects of other policies on multiple planetary boundaries.

Our analysis reveals that global carbon pricing, defined as a fee, tax, or restriction imposed on the burning of carbon-based fossil fuel sources, including coal, oil and gas (effectively raising the price on fossil fuels), can relieve pressure on all ESPs except land use and freshwater. Although a global carbon price is unlikely to be adopted, several national examples do exist[19]. Recent studies also suggest that a carbon price if combined with revenue recycling could receive public support[20]. We also consider the effect of reducing current subsidies for biofuel production as a complementary policy. Such a policy has at times been suggested[21] and is, in fact, implied by the EU Energy Directive that seeks to limit biofuel usage from food and feed crops. We find that the combination of carbon pricing and reduction in biofuel subsidies appears able to ease all of the planetary pressures.

To summarize, the case for a global carbon price appears to be even stronger in a multi-boundary world than when considering climate change as an isolated problem. Caution is however, warranted since higher carbon prices tend to make biofuel production more competitive, which implies that auxiliary policies will be needed in order to reduce all of the key planetary pressures outlined in refs. [2,3].

## Results

**Economic drivers of planetary pressures.** The planetary boundaries can be viewed as a list of the greatest global environmental problems caused by and facing mankind. Since these environmental problems are driven by economic activities, the first step in understanding the nature of the problem is to identify the principal sources of anthropogenic pressure on each ESP, and the links between them. Supplementary Table 1 summarises how specific activities in different economic sectors affect each ESP in the planetary boundaries framework, based on a review of the literature. It should be noted that our interpretations of the boundaries mostly follows[3], with the exceptions of the biosphere integrity and novel entity boundaries. These boundaries were replaced by the previous definitions, biodiversity loss, and chemical pollution found[2], as this greatly reduced the ambiguity in identifying the sources of pressure.

As is evident from Supplementary Table 1, and previously pointed out in e.g. refs. [22,23], the agricultural sector creates a substantial source of pressure on the ESPs. Hence, despite agricultural activity accounting for merely 4% of global economic output[24], it uses a very large share of the planets natural resources. About 40% of the Earth's land surface is used for agriculture, and it is still the primary driver of land conversion. It also contributes to roughly a quarter of global greenhouse gas emissions and accounts for over 90% of global freshwater, phosphorus, and nitrogen use. Agriculture is thus a primary driver of freshwater over-consumption and biogeochemical loading. Apart from the agricultural sector, fossil fuel consumption is also a key source of pressure on several ESPs. This is due partly to its direct impact on climate change, aerosol loading and ocean acidification and partly due to the pressure exerted on other ESPs indirectly through economic channels. For example, biogeochemical flows are highly dependent on fossil fuels. It is thus evident that both agriculture and fossil fuel consumption, as well as their interaction, are vital components of any model that aims to capture how human activity exerts pressure on the ESPs.

Further details and references which motivate the choice of economic sectors included in our model are provided in Supplementary Note 1.

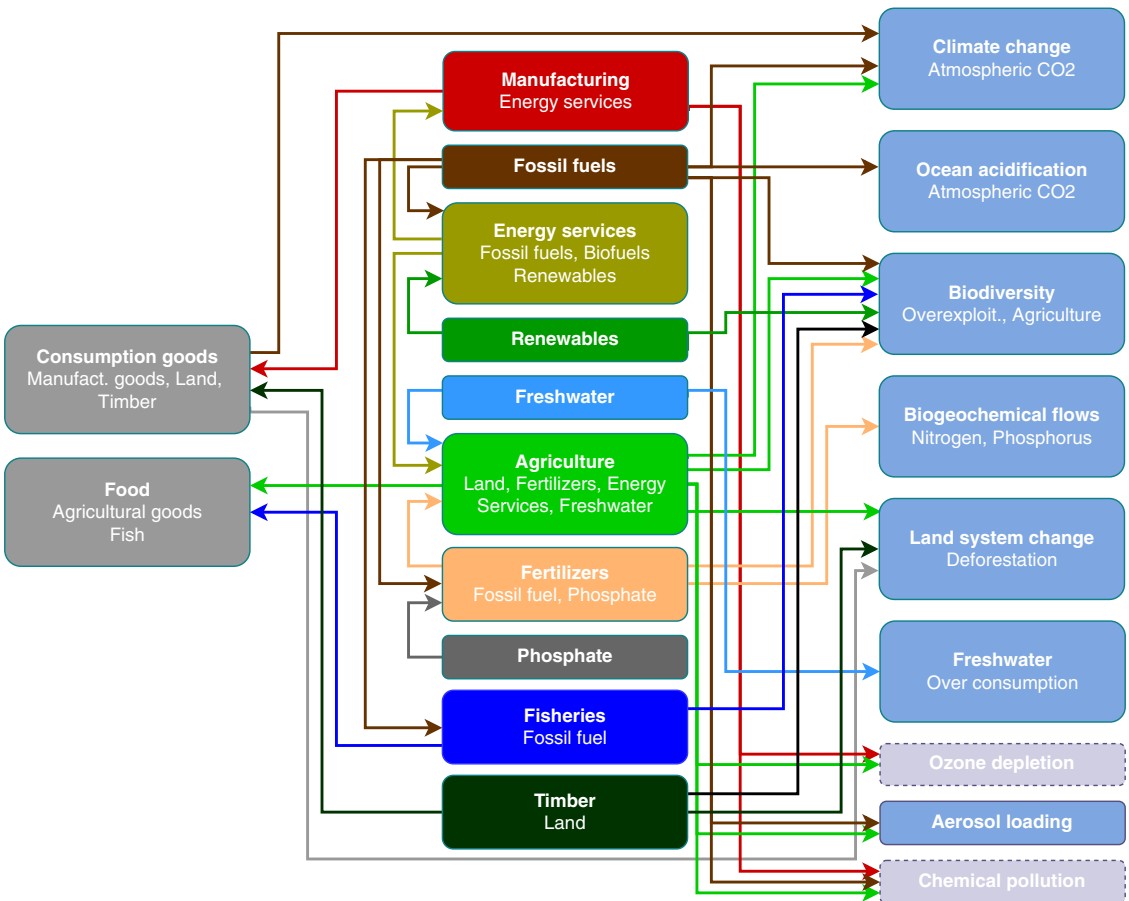

**Fig. 1 Schematic of the integrated economic-planetary boundaries model.** The above schematic gives an overview of the direct links existing in the model. The model is built in three layers (columns): consumption, production, and ESPs. The arrows indicate the direction of the economic inputs and outputs and which planetary processes they have an impact on. With the exception of ozone depletion and chemical pollution the impact of the two policy scenarios we consider are quantitatively assessed.

**Model description.** In order to systematically study the combined effects of economic linkages between the ESPs, we develop a model of the global economy with a focus on the key economic drivers of planetary pressure. Having a coherent economic framework that links the different ESPs via markets allows us to investigate whether a policy enacted in one domain is likely to create serious risks due to an increasing pressure imposed on other ESPs. We follow standard practice in economic modeling by characterizing a decentralized market structure, with a multitude of economic actors who maximize their individual objectives, resulting in competition for a limited set of resources, where the resulting allocation is often referred to as a competitive equilibrium. The model has many elements in common with integrated assessment models (IAMs), such as the DICE[10] or IMAGE[25] models, but also differs in many aspects, such as degree of detail, aggregation and solution approach adopted, primarily as a result of being intentionally developed to answer our specific research question.

Figure 1 provides a schematic outline of the model. Guided by the findings discussed in Supplementary Note 1, our model characterizes production choices in the following key economic sectors: agriculture, biofuels, timber production, fertilizer production, phosphate extraction, water extraction, manufacturing, energy services, fossil-fuel production, renewable energy (other than biofuels) and fisheries. In total, these sectors account for more than 90% of the drivers for the majority of the planetary pressures (see Supplementary Table 1 for a summary and Supplementary Note 1 for a detailed discussion). Many of the sectors are economically linked, which allows for an evaluation of many critical trade-offs, for instance, the allocation of land across agriculture and timber production (as in ref. [26]) or of agricultural output, which can be used as either food or biofuel (similar to ref. [27]). The model details are outlined in the "Methods" and Supplementary Methods section.

Two broad features of our modeling approach are however, worth highlighting. First, we approach computations via a method known as comparative statics, an approach that can be characterized as an analysis of the effects of exogenous policy changes using a linear approximation around the equilibrium outcome. Interventions like taxes and quotas can thus be represented as perturbations to the equilibrium, making it easy to trace out their propagation through the entire economy. The effects on the planetary pressures are computed as the net effects of the policy-induced changes in the various economic activities. Apart from increased transparency, this approach also has the advantage that the model parameters are easier to interpret (for instance, instead of more abstract production function parameters, the model uses shares of total input expenditures in a sector that goes to a specific input). In total, three sets of parameter estimates need to be assessed, elasticities (demand, supply, and substitution), expenditure shares of the economic sectors and finally quantity shares (see "Methods").

Second, our focus is on providing a qualitative understanding of the central processes and interactions involved. Hence, our modeling framework does not at present incorporate dynamic aspects of the problem. The model also does not include

feedbacks from the planetary pressures to the economy and human welfare. These feedbacks, although important, are unfortunately not understood nearly well enough to inform this kind of modeling exercise, and are thus left for future research.

In summary, our analysis is intended to capture the effects of changes in the economic environment in the short- to medium-term (perhaps 5–10 years). A longer-term analysis would need to take additional aspects, such as technological change and dynamic feedback already referred to, into account. One should be careful, therefore, in applying the framework developed here to long-run decision problems where dynamic feedbacks are likely to be important.

**Model parameterization**. Among the three broad set of parameter estimates described above, the majority of the estimates needed, can be expressed in terms of expenditure and quantity shares of goods going to different uses (see "Methods"). Detailed sectoral databases, such as the one underlying the widely-used GTAP model[15], allow us to derive a set of internally consistent parameter values. Our model's 39 parameters are a mix of expenditure shares, quantity shares, and elasticities, with the elasticities chosen to match empirical studies and the shares derived using data from the GTAP database. A complete list of parameters and their sources can be found in the "Methods", under the Expenditure shares section.

**Global carbon pricing**. We now consider the policy experiment of a marginal increase in the global carbon price and study its effect on the equilibrium outcome. In the model, the policy is implemented as a tax on fossil fuels (a carbon tax). The analysis would, however, be equivalent for any other policy involving pricing emissions from fossil fuels (e.g., a cap-and-trade system). As described above, the effects of a policy that increases the carbon price is calculated using a linear approximation around the equilibrium outcome and the derived numbers can be interpreted as percentage changes in response to a one percentage point change in the carbon price. Hence, it should be noted that results from larger perturbations should be interpreted based on the extent to which one can assume that a linear approximation constitutes a reasonable proxy for the true effect.

The direct effect of imposing this increase in the tax rate on fossil fuel use is clearly to increase its (after tax) price leading to a reduction in its demand. Our analysis shows that a one percentage point increase in the global carbon tax rate would reduce annual global $CO_2$ emissions by 0.25% (equivalent to 0.11 $GtCO_2$ $yr^{-1}$), and emissions from fossil fuels by 0.36% (see Supplementary Table 2). The implied elasticity of emission reductions is smaller than in some model-based analyses but is relatively well-aligned with empirical estimates (see Supplementary Note 2).

This reduction in emissions ameliorates pressure on both climate change and ocean acidification (depicted as arrows at the top and bottom of Fig. 2, pointing inward toward the safe operating space). This change does not come from a single sector, but rather is the net effect of changes in fossil fuel use in the production of energy services, fertilizer, land use change, etc. Carbon dioxide emissions from fossil fuel use decrease in most sectors as a result of the tax, while emissions from land use change increase due to substitution toward land in agriculture and increased biofuel production. Similar direct impacts occur with aerosol loading. The net effect is a reduction in aerosol loading (measured in terms of aerosol optical depth) following a reduction in the release of atmospheric particles from fossil-fuel burning.

As producers and consumers of fossil fuel react to changing prices, second-order effects arise. To start with, an increase in the carbon tax increases the cost of the nitrogen component of

fertilizers, since nitrogen fixing uses a very fossil-fuel intensive industrial process. Since nitrogen has a high degree of complementarity with phosphate in the production of fertilizers (most fertilizers are sold as multi-nutrient mixtures), demand for phosphate also decreases, reducing the overall pressure on biogeochemical flows. Furthermore, a carbon tax turns out to reduce pressure on our measure of biodiversity. This is partially due to reduced activity in key sectors such as agriculture and fisheries, in which fossil fuel is an important input. The total effect on biodiversity is the net result of a number of different effects, some positive others negative (see "Numerical results" in "Methods", for details). Chemical pollution and stratospheric ozone depletion are only qualitatively assessed based on their relationship to the model variables. If all related model variables move in a direction that decreases the pressure, we draw the conclusion that the net effect is reduced pressure. This turns out to be the case for chemical pollution. For stratospheric ozone depletion, however, the effects go in different directions and we cannot determine the net direction.

A carbon tax will also increase the pressure on land-system change, a net result of several opposing effects. First, a higher relative price of non-land inputs (energy and fertilizer) will encourage substitution toward greater land use in the agricultural sector. At the same time, the higher price of fossil fuel raises the relative price of manufacturing goods compared to e.g., timber and recreation. The demand for non-agricultural uses of land thus counteracts the increase in demand from agriculture. Together with land conversion costs, this results in a small overall increase in land use in the agricultural and timber sectors, which comes at the expense of natural land. Finally, a carbon tax leads to a small increase in freshwater use. In this case, the increased freshwater use is primarily due to substitution away from more expensive agricultural inputs. The effect of the carbon tax upon freshwater use illustrates the implications of global aggregation in modeling, i.e., our global aggregate model responds with increased aggregate water use. In reality, with not all farms across the world being able to substitute freshwater for energy-intensive inputs (e.g., in subsistence farming with no irrigation), other inputs or output must adjust. If freshwater use is constrained in this way, a carbon tax may end up driving down output, or encourage greater substitution along other margins that might exacerbate other planetary pressures.

It is evident that direct effects (on climate change and ocean acidification) are significantly larger than the indirect. The effect on nitrogen use, which is smaller but of similar magnitude to that of climate change, is close to direct, since fossil fuel is an important input in its production. The remaining effects are more indirect and they are an order of magnitude smaller. However, if we were to scale up the effects to the size of the carbon tax required to meet climate policy targets, these smaller indirect effects would be significant.

An increase in the global carbon tax thus reduces pressure on many ESPs besides climate change. The argument for a carbon tax is typically made considering only climate change. Our analysis suggests that in a richer framework, that considers multiple planetary boundaries, the case for a carbon tax is in some ways even stronger. This richer model, however, also alerts us to certain risks, and next, we thus investigate whether a complementary biofuel policy could avert these dangers and move us more firmly toward the safe operating space.

**Reduction of biofuel subsidies**. A global carbon tax exerts pressure on the land-system and freshwater ESPs mainly because

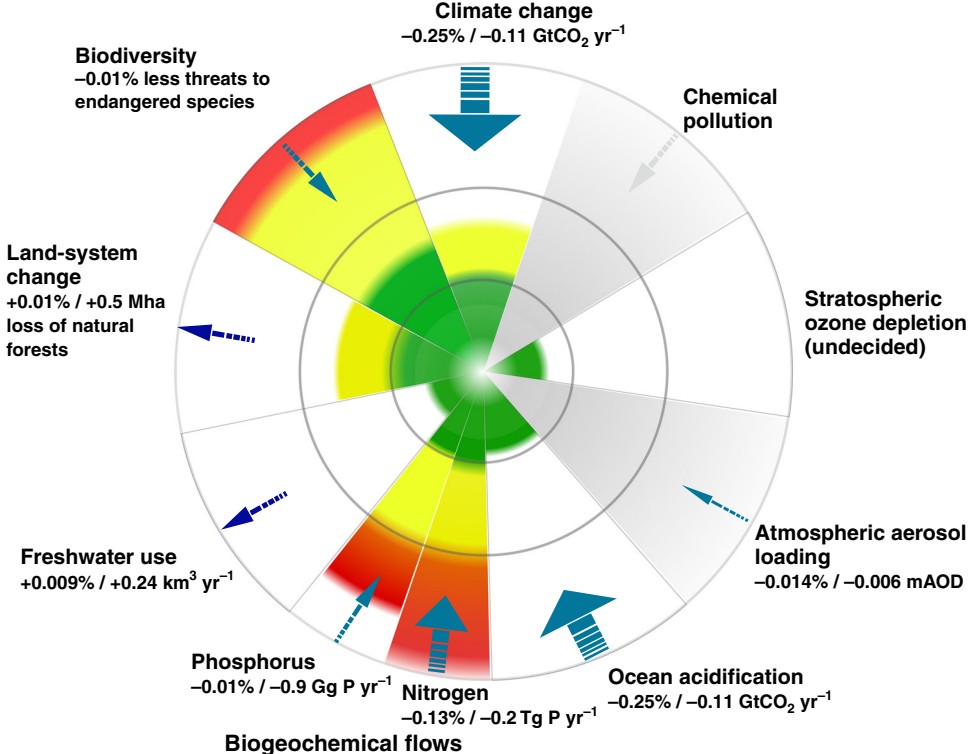

**Fig. 2 Changes in planetary pressures resulting from a one percentage point increase in the carbon tax.** This figure is a modification of the original planetary boundary figure from refs. [1,3]. The colors indicate the current state for each boundary: green, yellow, and red correspond to safe, increasing risk and high risk, respectively. We have added arrows illustrating the effects on each individual ESP, from increasing the carbon tax rate by one percentage point in our integrated model of the global economy and the ESPs. The direction of the arrows indicate increasing or decreasing pressure, while the width of the arrows are indicative of the magnitude of change. For chemical pollution and stratospheric ozone depletion, we only derived the qualitative direction of change. Further details are given in Supplementary Table 2.

it increases the demand for agricultural output. Delving deeper into the sectoral linkages, one can see that this is not chiefly driven by demand for food, but rather demand for biofuels (which is a substitute for fossil fuel). This suggests, that in order to ease the remaining planetary pressures, we need to complement the carbon tax with some additional policy limiting the demand for biofuels.

We thus consider the effect of a complementary reduction of biofuel subsidies (implemented as an increase in a biofuel tax). Biofuel production is currently heavily subsidised in large parts of the world[28]. The question is whether it would be prudent to scale back these subsidies in a world with a higher carbon tax. Fig. 3 summarizes the net changes in our model resulting from this two-pronged policy: a one percentage point increase in the carbon tax rate and a one percentage point reduction of the subsidy rate for biofuels.

As is evident from the figure, this combination of policies can ameliorate pressures on all the ESPs. Biofuel production has not only a negative climate-related effect from increased land use, but also a negative effect on both biogeochemical flows and freshwater systems due to the increased demand for fertilizer and freshwater. Similar results have also been found by ref. [29] in an assessment of the consequences of a large-scale deployment of bioenergy with carbon capture and storage as a measure for mitigating climate change. Importantly, this result does not necessarily imply that a biofuels subsidy is a bad idea in the absence of a carbon tax. Such a policy would reduce some pressures while increasing others. This analysis assumes biofuels are produced using the same inputs as food and feed, reflecting current production patterns (with biofuel production using between 1 and 3% of total cropland area, see ref. [21]). When biofuels of this type are phased out in favor of those not competing directly with food crops for land, either by policy (as required by a new EU Renewable Directive) or technology change (so-called second-generation biofuels), then biofuels may be considered instead as one among other renewable energy sources like solar and wind power.

**Sensitivity analysis.** To assess the degree to which our findings are sensitive to parameter choices, we identify key parameters regarding which uncertainty is greater, and choose a range within which they vary. We then solve the model for all possible combinations of the lower and upper bounds for these parameters, recording the maximum and minimum predicted changes for all model variables.

When a carbon tax is the only policy considered, the signs of the changes in planetary pressures are mostly unaffected, though there are a few notable sign changes. In some extreme scenarios, land use in agriculture and freshwater use may both decrease, and in rare cases, we also observe an increase in phosphate use. Overall, our sensitivity analysis suggests that, if anything, the outcome distribution tends to be skewed toward reduced rather than increased planetary pressures.

When we supplement the carbon tax with a complementary biofuel subsidy reduction, our results are not as sensitive. There is no reversal in the sign of the net effects. The only change of any relevance for our analysis is that we find an increase in food production from agriculture. The details of the sensitivity analysis are reported in Supplementary Table 2.

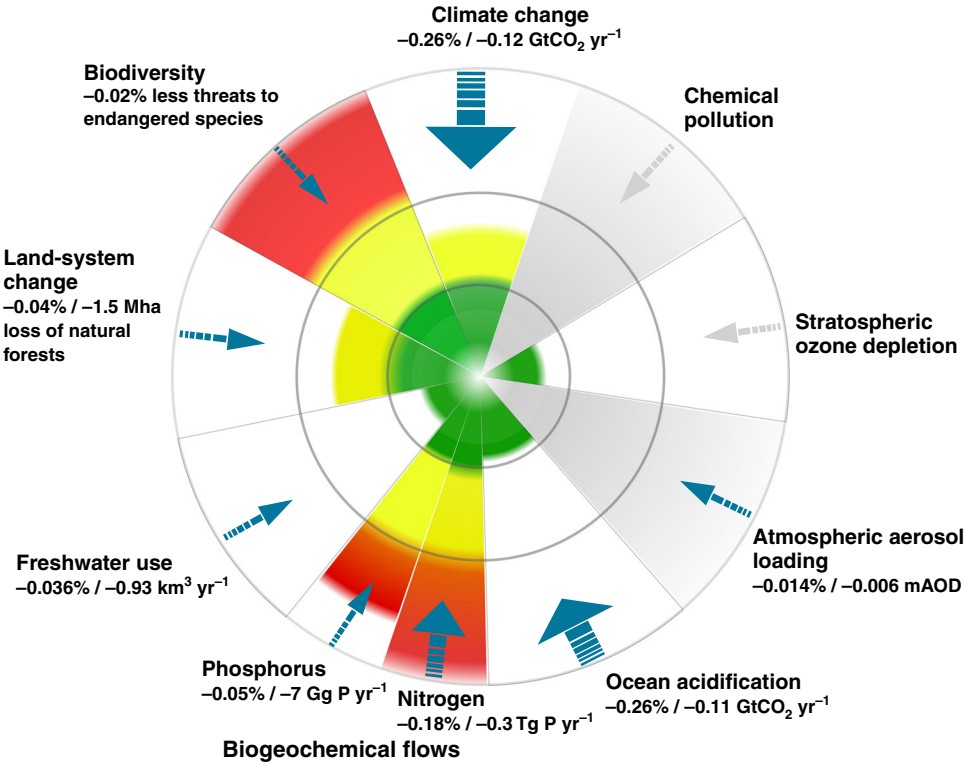

**Fig. 3 Change in planetary pressures resulting from a one percentage point increase in the tax on carbon and a one percentage point reduction of biofuel subsidies.** This figure is a modification of the original planetary boundary figure from refs. [1,3]. The colors indicate the current state for each boundary: green, yellow, and red correspond to safe, increasing risk and high risk, respectively. We have added arrows illustrating the effects on each individual ESP, from increasing the tax rate on carbon by one percentage point and reducing biofuel subsidies by one percentage point in our integrated model of the global economy and the ESPs. Interpretation is otherwise the same as in Fig. 2. Further details are given in Supplementary Table 3.

## Discussion

Carbon pricing is typically justified purely on the basis that it helps mitigate climate change. The planetary boundaries framework developed by Rockström and colleagues[1] begs the question as to whether carbon pricing could have unintended side effects, stabilizing or destabilizing, other environmental processes. Here we present an integrated analysis showing that a higher global carbon price may be sufficient to single-handedly reduce almost all planetary pressures. Some of the effects are direct and easy to anticipate, since fossil fuel consumption exerts an important pressure on several ESPs, including climate change and ocean acidification. Other consequences are more indirect, such as a reduction in nutrient-loading, which derives from the importance of fossil fuels in nitrogen production and complementarity between phosphate and nitrogen in fertilizer production. Our analysis shows that, while these indirect effects are significantly smaller than the direct effects, they are still far from insignificant.

A carbon price can also indirectly increase other planetary pressures, especially through increased demand for agricultural land resulting from the increased biofuel demand due to higher fossil fuel prices. We show that a complementary policy of scaling back biofuel subsidies as the carbon price is increased would help avoid these negative effects. This combination of policies provides a means of reducing all planetary pressures.

When interpreting these results, it is however important to bear in mind that we do not make an assessment of the welfare consequences of these changes, meaning that the relative size of the effects need not translate directly to the relative magnitude of resulting welfare effects. Further, while we considered two specific policies in this paper, the model can very easily be adapted to study the effects of a wide variety of policies. For future research,

we anticipate extending the framework developed here in many directions, accommodating aspects such as dynamics, uncertainty, and welfare analyses.

To summarize, our results suggest that carbon pricing in combination with a reduction in biofuel subsidies can alleviate all key planetary pressures outlined in the planetary boundary framework, suggesting that the case for a global carbon price appears even stronger in a multi-boundary world than when considering climate change in isolation.

## Methods

**Model components**. The results of this paper are derived from a model that is built around the economic sectors outlined as the most important drivers of planetary pressures in Supplementary Table 1. This includes production sectors that have an important direct effect on the ESPs or that have important links to such sectors. They may be linked by using output from such sectors as inputs, providing inputs to such sectors, competing for inputs with such sectors or providing outputs that serve as substitutes for the output from those sectors. The resulting set of included production sectors are: agriculture (producing food and biofuel), energy services, fossil-fuel extraction, renewable energy (other than biofuel), fertilizer production, phosphate extraction, water supply, fisheries, and industrial manufacturing. The demand for final consumption goods is derived from the maximization of households' utility. Since we have economic policies in the model, we are implicitly assuming some government entity that imposes these policies, but since we consider the policies exogenous (not, e.g., determined to optimize some objective) we do not explicitly model the government.

We solve the model as a competitive equilibrium where we assume that all agents maximize their respective objectives while taking prices as given (prices are given from the perspective of the individual agent, but are endogenously determined by aggregate supply and demand). We then analyze changes in the endogenously determined model variables in response to an assumed exogenous change in economic policy.

In the model, competition for resources thus leads to a number of important trade-offs. These arise from three main sources including, alternative uses of the output of a sector (e.g., output from the agricultural sector can be used as food or

**Table 1 Model quantities, prices and uses.**

| Variable | Quantity | Price | Uses |
|---|---|---|---|
| $A$ | Agricultural production | $p_A$ | Food $A_\mathcal{F}$, biofuels $A_B$ |
| $E$ | Fossil fuel | $p_E$ | Energy $E_\mathcal{E}$, Fisheries $E_F$ and fertilizer production $E_P$ |
| $\mathcal{E}$ | Energy services | $p_\mathcal{E}$ | Agriculture $\mathcal{E}_A$, manufacturing $\mathcal{E}_Y$ |
| $F$ | Fisheries | $p_F$ | Food |
| $L$ | Land | $p_L$ | Agriculture $L_A$, timber production $L_T$, natural land $L_U$ |
| $M$ | Other inputs | $p_{M_X}$ | $M_X$, for $X \in \{A, F, P, T, Y\}$ |
| $P$ | Fertilizers | $p_P$ | Agriculture |
| $\mathcal{P}$ | Phosphate | $p_\mathcal{P}$ | Fertilizer production |
| $R$ | Renewables (excluding biofuels) | $p_R$ | Energy services |
| $T$ | Timber production | $p_T$ | Consumption |
| $W$ | Fresh water | $p_W$ | Agriculture |
| $Y$ | Manufacturing | $p_Y$ | Consumption |

biofuels), sectors competing for the use of inputs (e.g., land can be used for agriculture, forestry or maintained as undisturbed natural land) or from inputs being substitutes or complements in production or consumption (e.g., nitrogen and phosphorus preferably being used in fixed proportions).

The production sectors are modeled either by using an explicit production function or by a production cost function. A production function is specified for agriculture, energy services, fertilizer production, fisheries, timber production and industrial manufacturing sectors since their factor inputs are directly connected to one or more ESPs (see the previous section on "Economic drivers of planetary pressures"), thus making their input substitutability important. For all sectors except agriculture, we use one level constant elasticity of substitution (CES) functions. For agriculture, we use a nested CES function (see below). Sectors whose production processes are of less importance, are represented by a production cost function. These sectors include phosphate, water, fossil fuel, and renewable energy. Also, in many sectors, certain inputs e.g., labor and capital, are economically important but their explicit modeling is not directly relevant for our analysis (i.e., of negligible importance to the ESPs). To account for these inputs, we include an aggregate input, which we refer to as other inputs, in all production sectors except energy services and assume that these are supplied with a given sector-specific price elasticity of supply. The possibility of adjusting these other inputs leads to decreased use in sectors where their marginal value decreases and increased use in sectors where their marginal value increases, and thus to some extent captures the possibility to move inputs between sectors in response to changing economic conditions.

We will now present the model sectors in more detail. A list of model quantities, their prices and uses can be found in Table 1 (different uses of a quantity are denoted by subscripts).

The agricultural sector uses inputs land ($L_A$), fertilizers ($P$), water ($W$), energy services ($\mathcal{E}_A$) and other inputs ($M_A$) as inputs to produce output that can be used for food or biofuels. Producers maximize their profit, taking prices as given. Their profit maximization problem is

$$\max_{L_A, P, W, \mathcal{E}_A, M_A} p_A A(L_A, P, W, \mathcal{E}_A, M_A) - p_L c_A(L_A) L_A \\ - p_P P - p_W W - p_\mathcal{E} \mathcal{E}_A - p_{M_A} M_A, \quad (1)$$

where $c_A(L_A)$ captures the cost of converting land to agricultural land. The agricultural production function is a CES function between land and non-land inputs, where non-land inputs are aggregated using a CES function.

The energy-services sector combines energy from different sources into a bundle of energy services ($\mathcal{E}$). The different sources are biofuels ($A_B$), fossil fuels ($E_\mathcal{E}$) and renewables ($R$). The producers in this sector solve the profit maximization problem

$$\max_{A_B, E_\mathcal{E}, R} p_\mathcal{E} \mathcal{E}(A_B, E_\mathcal{E}, R) - p_A A_B - p_E E_\mathcal{E} - p_R R. \quad (2)$$

We model production of fertilizers ($P$) as using fossil fuel ($E_P$), phosphate ($\mathcal{P}$) and other inputs ($M_P$). The use of fossil fuel is intended to capture the fossil-fuel (more specifically natural-gas) intensive production of the nitrogen component of fertilizers. We thus treat fossil fuel use in fertilizer production as a proxy for nitrogen. The profit maximization problem of fertilizer producers is

$$\max_{E_P, \mathcal{P}, M_P} p_P P(E_P, \mathcal{P}, M_P) - p_E E_P - p_\mathcal{P} \mathcal{P} - p_{M_P} M_P. \quad (3)$$

For timber production ($T$) we only consider the input land ($L_T$) and other inputs ($M_T$). The producers then solve the maximization problem

$$\max_{L_T, M_T} p_T T(L_T, M_T) - p_L c_T(L_T) L_T - p_{M_T} M_T, \quad (4)$$

where $c_T$ is a cost of converting (e.g., clearing) land for forestry.

Industrial manufacturing ($Y$) requires energy ($\mathcal{E}_Y$) and other inputs ($M_Y$). While we refer to this sector as manufacturing, the substitutability between energy and other inputs is chosen to match that of the economy as a whole. The substitutability thus reflects not only the manufacturing sector but also the service sector that has a significantly lower energy intensity but is economically important. The maximization problem of the representative producer is

$$\max p_Y Y(\mathcal{E}_Y, M_Y) - p_\mathcal{E} \mathcal{E}_Y - p_{M_Y} M_Y. \quad (5)$$

The fisheries sector uses inputs fossil fuel ($E_F$) and other inputs ($M_F$). The producers solve the maximization problem

$$\max_{E_F, M_F} p_F F(E_F, M_F) - p_E E_F - p_{M_F} M_F. \quad (6)$$

Extraction of fossil fuel ($E$) is modeled by assuming a gross extraction cost ($g_E$) that increases with increased extraction ($g_E(E)$ thus gives the total cost of extracting quantity $E$). We assume that the tax on fossil fuels (a percentage tax $\tau_E$) is paid by the firms that extract and sell it. Extraction firms solve the profit maximization problem

$$\max_E \frac{p_E}{1 + \tau_E} E - g_E(E). \quad (7)$$

The sectors phosphate ($\mathcal{P}$), water ($W$), renewable energy (other than biofuels) ($R$) and the other inputs ($M_A$, $M_F$, $M_P$, $M_T$, and $M_Y$) are similarly represented by a production or extraction cost and the profit-maximization problem of the producers are given by

$$\max_X p_X X - g_X(X) \text{ for } X \in \{\mathcal{P}, W, R, M_A, M_F, M_P, M_T, M_Y\}. \quad (8)$$

We have now described the maximization problems underlying decisions made by all producers. The representative household also solves a maximization problem, maximizing the utility derived from consumption. The households' preferences are represented by utility function $U$ and the utility-maximization problem, subject to the income being $I$, is given by

$$\max_{A_\mathcal{F}, F, Y, L_U, T} U\big(\mathcal{F}(A_\mathcal{F}, F), \tilde{\mathcal{F}}(Y, L_U, T)\big) \quad (9)$$

$$\text{s.t. } p_A A_\mathcal{F} + p_F F + p_Y Y + p_L L_U + p_T T \le I.$$

This specification has divided consumption into two levels. While this division is not necessary at this level of generality, it clarifies the assumed substitutabilities between goods. We assume greater substitutability within than between categories. The upper level consists of food ($\mathcal{F}$) and non-food ($\tilde{\mathcal{F}}$) goods, with the former category consisting of food from agriculture and from fisheries, and the latter of manufactured goods, natural land and timber. The inclusion of natural land is intended to capture various ways in which households' demand for natural lands lead to land being kept from other uses, e.g., preservation of land as national parks. We assume that timber is consumed directly by the households.

This completes the description of the modeling of all decision-making agents in the model. In addition to conditions derived from these maximization problems, we must also specify market-clearing conditions that make sure that supplied and demanded quantities add up.

For land ($L$), the total supply is assumed to be fixed:

$$L = L_A + L_T + L_U. \quad (10)$$

The remaining market-clearing conditions are for agricultural production

$$A = A_\mathcal{F} + A_B, \quad (11)$$

fossil fuel

$$E = E_\mathcal{E} + E_F + E_P \quad (12)$$

and energy services

$$\mathcal{E} = \mathcal{E}_A + \mathcal{E}_Y. \quad (13)$$

In summary, production functions, market-clearing conditions, budget constraints and first-order conditions from the maximization problems of representative agents provide us with 41 equilibrium conditions pinning down the 41 endogenous prices and quantities. The full set of equilibrium conditions are available in the Supplementary Methods.

**Solution Approach**. We note a few features of our model, some of which have already been mentioned: there are no explicit externalities; policies are applied exogenously; all sectors are assumed to be competitive; market clearing determines the equilibrium. In this context, we can work with the decentralized equilibrium, which may be analyzed by considering the first order conditions. In our model, there are 41 unknown prices and quantities in the model, determined by 41 equilibrium conditions. Being exogenous, policies represent parameters that are

**Table 2 Parameters—quantity shares.**

| Parameter | Source | Value |
|---|---|---|
| $Q_{L,L_A}$ | Source:[47] | 53.0% |
| $Q_{L,L_T}$ | Source:[47] | 2.0% |
| $Q_{A,A_B}$ | Derived in section "Quantity shares". | 3.8% |
| $Q_{\mathcal{E},\mathcal{E}_A}$ | Derived in section "Quantity shares". | 5.0% |
| $Q_{E,E_P}$ | Derived in section "Quantity shares". | 1.4% |
| $Q_{E,E_F}$ | Derived in section "Quantity shares". | 0.4% |

**Table 3 Parameters: expenditure shares (source: GTAP).**

| Expenditure share | Value |
|---|---|
| $\Gamma^A_{L_A}$ | 19.2% |
| $\Gamma^{\bar{L}_A}_{P}$ | 8.0% |
| $\Gamma^{\bar{L}_A}_{W}$ | 2.4% |
| $\Gamma^{\bar{L}_A}_{\mathcal{E}_A}$ | 4.1% |
| $\Gamma^{\mathcal{E}}_{A_B}$ | 0.4% |
| $\Gamma^{\mathcal{E}}_{E_{\mathcal{E}}}$ | 94.3% |
| $\Gamma^{U}_{\mathcal{F}}$ | 12.3% |
| $\Gamma^{\mathcal{F}}_{F_{\mathcal{E}}}$ | 3.4% |
| $\Gamma^{\mathcal{F}}_{Y}$ | 99.1% |
| $\Gamma^{\mathcal{F}}_{L_U}$ | 1.7% |
| $\Gamma^{T}_{L_T}$ | 37.5% |
| $\Gamma^{Y}_{\mathcal{E}_Y}$ | 6.4% |
| $\Gamma^{P}_{E_P}$ | 10.9% |
| $\Gamma^{P}_{\mathcal{P}}$ | 31.3% |
| $\Gamma^{\bar{F}}_{E_F}$ | 22.8% |

known in advance; denote a generic "policy" pertaining to any one ESP by $\tau$. Let $X_i$ denote the generic $i$th variable, an endogenous price or quantity. The $j$th equilibrium condition can then generally be written as:

$$G_j(X_1, \ldots, X_{41}; \tau) = 0. \tag{14}$$

This system of equations implicitly define all resulting equilibrium quantities and prices as functions of the policy i.e. $X_i = X_i(\tau)$.

There are now two solution approaches: the first is to solve the set of resulting non-linear equations (and thereby obtain all the equilibrium values); the second is to trace out marginal changes in the equilibrium values in response to a change in the policy, $\tau$. The latter approach can be illustrated by considering the total derivative of the equilibrium conditions with respect to the policy. This leads to a system of equations, with the $j$th equation being

$$\sum_i^{41} \left[ \frac{X_i}{G_j} \frac{\partial G_j}{\partial X_i} \hat{X}_i \right] = -\frac{1}{G_j} \frac{\partial G_j}{\partial \tau}, \tag{15}$$

where

$$\hat{X}_i \equiv \frac{1}{X_i} \frac{dX_i}{d\tau} \tag{16}$$

is the relative change in variable $X_i$. These can be interpreted as a linear approximation of the percentage change in the variable induced by a one percentage point increase in the fossil fuel tax. Assume, for instance, that we get $\hat{X}_i = 2$ and consider a one percentage point increase in the tax rate, $\Delta\tau_E = 0.01$. We would then get $\frac{1}{X_i}\Delta X_i \approx \hat{X}_i \Delta\tau_E = 0.02$. Hence, a one percentage point increase in the tax induces a two percent increase in the quantity. The result is a system of 41 equations in 41 unknowns, the $\hat{X}_i$, and is most useful because of linearity in the unknowns. Indeed this approach can be viewed as linear approximation of the equilibrium response to a change in the policy parameter. The required empirical parameter values needed for numerical computations are fewer, easier to find, and easier to interpret. Furthermore, if considering changes in other parameters of the model (e.g., changes in other policies) only the right-hand side of (15) needs to be changed.

**Data and parametrization**. We parameterize the model based partly on data extracted directly from the widely-used GTAP database, described below, and partly on empirical estimates from various sources in the literature. As described above, we mainly need three types of values: quantity shares, expenditure shares and elasticities of various kinds. In total, we need 39 empirical estimates to run the model. In our computations, we set the initial carbon price equal to zero. In reality there are various forms of carbon prices. It is difficult to get a precise measure of all these, but the global average is likely a relatively small negative price. For our analysis, this makes little difference. Assuming a different initial price would scale all results somewhat since the effect of a one percentage point increase in the price would, relatively speaking, be smaller or larger depending on the initial price. All other parameter values that we use are empirically derived based on the current effective carbon price. In the following section, we provide tables with parameter values and their sources.

The first type of parameter that occurs are quantity shares. By quantity share $Q_{X,X_Z}$ we mean the share of total quantity $X$ used in a specific sector $Z$. The full set of values, including their sources are given in Table 2. The exceptions are the quantity shares of fossil fuel going to different sectors and the share of agricultural production going to food or biofuel. These were derived as follows.

Total energy consumption in 2011 was 12,225 Mtoe[30]. Out of this, 10624 Mtoe came from fossil fuel related sources. Fertilizer production uses about 1.2% of total energy supply and almost all of this comes from fossil fuels[31]. Hence we assume that the share of fossil fuels going to fertilizer production is $Q_{E,E_P} = \frac{12,225}{10,624} \times 1.2\% \approx 1.4\%$. For fisheries production, we assume a global fuel consumption of 40 billion litre's of fuel[32]. Assuming that this is mostly diesel, this corresponds to 40 Mtoe of fossil fuel or $Q_{E,E_F} = \frac{40}{10,624} \approx 0.4\%$ of total fossil fuel use. Finally we assume the remaining fossil fuels are used in energy production i.e., 98.2%.

In order to compute the share of agricultural production going to biofuels we used data underlying the FAO Agricultural Outlook report 2016–2025[33]. For each major agricultural commodity (e.g., wheat, maize, rice, etc.) we computed the share of agricultural production used for biofuels and then computed a weighted sum using the fraction of land used to harvest a specific commodity as weight. This resulted in a quantity share $Q_{A,A_B} \approx 3.8\%$.

Agriculture accounts for only a relatively small proportion of total final energy demand in both industrialized and developing countries. In OECD countries, for example, around 3–5% of total final energy consumption is used directly in the agriculture sector, while for developing countries, the equivalent figure is likely slightly higher in the range of 4–8% of total final commercial energy use[34]. Based on these estimates, we concluded that $Q_{\mathcal{E},\mathcal{E}_A} = 5\%$ constitutes a reasonable baseline.

The second type of that occurs in our equilibrium conditions are expenditure shares. The expenditure share $\Gamma^Z_X$ of input $X$ in sector $Z$ is the share of total spending on inputs in sector $Z$ that goes to $X$. To pin down these at the global level, we employed the GTAP database[15]. More specifically, we used the GTAP data set corresponding to the year 2014, for 141 countries and 57 sectors. The GTAP database is a unique global economic data set constructed by collating and reconciling data on national input-output tables, international trade, production, consumption, and macro-economic data sets from various international data sources. This has further been extended by ref. [35] to include renewable energy commodities, based on several energy data sources, including the International Energy Agency (IEA) data set and the World Bank data set. Furthermore, ref. [36] has extended this even further to include water as an endowment, in both agricultural and other sectors. Finally, we have a data set in which we can derive the shares of labor, capital, land, water, and several other inputs in producing all commodities. Some inputs, such as fertilizers are not separately identified in this data set, but they are subsumed in broader GTAP sectors such as chemicals, rubber, and plastics. Therefore, we make broad reasonable assumptions to derive the shares of such granular-level inputs; for example, we assume that most of agricultural consumption of output from the GTAP sectors chemicals, rubber, and plastics are fertilizers and pesticides. For all production sectors except energy services, we assign the residual expenditure share, remaining when all inputs of direct interest have been accounted for, to other inputs $M$. The details are given below and summarized in Table 3.

*Agriculture*. Our agricultural production function distinguishes between land and non-land inputs (with "other inputs" in the non-land category). The expenditure share of land is 19.2%. The expenditure shares of fertilizers, water, energy, and other inputs are 6.43%, 1.93%, 3.33%, and 71.1%, respectively. Their respective shares out of non-land inputs are their total shares divided by the total non-land share. This means that $\Gamma^A_{L_A} = 0.192$, $\Gamma^A_{\bar{L}_A} = 0.808$, $\Gamma^{\bar{L}_A}_{P} = \frac{0.0643}{0.808} = 0.0796$, $\Gamma^{\bar{L}_A}_{W} = \frac{0.0193}{0.808} = 0.0239$, $\Gamma^{\bar{L}_A}_{\mathcal{E}_A} = \frac{0.0333}{0.808} = 0.0412$, and $\Gamma^{\bar{L}_A}_{M_A} = \frac{0.711}{0.808} = 0.880$.

*Energy services*. The expenditure shares of biofuels, fossil fuels and renewables are 0.37%, 94.33%, and 5.30% respectively. That is $\Gamma^{\mathcal{E}}_{A_B} = 0.0037$, $\Gamma^{\mathcal{E}}_{E_{\mathcal{E}}} = 0.9433$, and $\Gamma^{\mathcal{E}}_R = 0.0530$.

Utility. The expenditure shares of food from agriculture, fish, manufactured goods, recreational land use, and timber are 11.93%, 0.42%, 86.86%, 0.15%, and 0.65%. This gives expenditure share of food $\Gamma^U_{\mathcal{F}} = 0.1235$ and expenditure share of non-food goods $\Gamma^U_{\bar{\mathcal{F}}} = 0.8765$. The within-category expenditure shares are $\Gamma^{\mathcal{F}}_{A_{\mathcal{F}}} = \frac{11.93}{12.35} = 0.9660$, $\Gamma^{\mathcal{F}}_{F} = \frac{0.42}{12.35} = 0.0340$, $\Gamma^{\bar{\mathcal{F}}}_{Y} = \frac{86.86}{87.65} = 0.9910$, $\Gamma^{\bar{\mathcal{F}}}_{L_U} = \frac{0.15}{87.65} = 0.001711$, and $\Gamma^{\bar{\mathcal{F}}}_{T} = \frac{0.65}{87.65} = 0.007416$.

*Timber*. The expenditure shares of land and other inputs are 37.48% and 62.52%, respectively. That is $\Gamma^T_{L_T} = 0.3748$ and $\Gamma^T_{M_T} = 0.6252$.

## Table 4 Parameters—elasticities and quantities.

| Parameter | Source | Value |
|---|---|---|
| $\sigma_U$ | Ref. [48] | [0.4, 0.6, 0.5] |
| $\sigma_{\mathcal{F}}$ | Ref. [49] | [1.13, 1.33, 1.23] |
| $\sigma_F$ | Standard values for extractive sectors used in GTAP[50] | [0.1, 1, 0.2] |
| $\sigma_T$ | Standard values for extractive sectors used in GTAP[50] | [0.1, 1, 0.2] |
| $\sigma_{\tilde{\mathcal{F}}}$ | Drawn from estimates of substitutability derived from ref. [51] | [1.5, 2.1, 1.8] |
| $\sigma_A$ | Ref. [26] | [1.1, 1.24, 1.14] |
| $\sigma_P$ | Literature suggests high complementarity, see e.g.,[52] | [0.05, 0.3, 0.2] |
| $\sigma_{\tilde{L}_A}$ | Assumed based on reading of the literature | [0.25, 0.75, 0.5] |
| $\sigma_{\mathcal{E}}$ | Ref. [53] | [1.5, 2.1, 1.8] |
| $\sigma_Y$ | Based on Table 2 in ref. [54] | [0.1, 1, 0.5] |
| $\Lambda_R$ | ref. [55] | 1/2.7 |
| $\Lambda_E$ | Based on estimates in ref. [56] | [0.8, 1.2, 1] |
| $\Lambda_W$ | Based on ref. [57] | 1/1.79 |
| $\Lambda_{\mathcal{P}}$ | Assumed based on reading of the literature | 1/1.5 |
| $\Lambda_{M_X}$ | Assumed with wide span for robustness | [0, 2, 1] |
| $V_T$ | Based on ref. [26] | 0.05 |
| $V_A$ | Based on ref. [26] | 0.05 |
| $\tau_E$ | A global carbon tax currently does not exist | 0.0 |

The values are depicted as [min, max, and mean] where the min and max values are used in the sensitivity analysis, while the mean values are used in the baseline simulation.

## Table 5 Baseline results.

| Variable | Carbon tax | | Carbon Tax + Biofuel policy | |
|---|---|---|---|---|
| | Quantity | Price | Quantity | Price |
| *Agricultural sector: production* | | | | |
| Total | −0.003 | 0.034 | −0.045 | −0.0 |
| Biofuels | 0.717 | 0.034 | −1.008 | −0.0 |
| Food | −0.032 | 0.034 | −0.007 | −0.0 |
| *Agricultural sector: inputs* | | | | |
| Land-share agriculture | 0.012 | 0.02 | −0.037 | −0.005 |
| Energy in agriculture | −0.293 | 0.609 | −0.355 | 0.618 |
| Fertilizer production | −0.021 | 0.065 | −0.062 | 0.033 |
| Water production | 0.009 | 0.005 | −0.036 | −0.02 |
| *Energy-related sectors and services* | | | | |
| Fossil-fuel in energy services | −0.368 | 0.636 | −0.362 | 0.641 |
| Fossil-fuel in fertilizer prod. | −0.135 | 0.636 | −0.184 | 0.641 |
| Fossil-fuel in fisheries | −0.211 | 0.636 | −0.223 | 0.641 |
| Energy services | −0.319 | 0.609 | −0.32 | 0.618 |
| Energy in manufacturing | −0.321 | 0.609 | −0.318 | 0.618 |
| Renewables production | 0.466 | 0.173 | 0.475 | 0.176 |
| *Extractive sectors* | | | | |
| Fossil-fuel extraction | −0.364 | 0.636 | −0.359 | 0.641 |
| Phosphate extraction | −0.007 | −0.005 | −0.049 | −0.033 |
| *Other* | | | | |
| Land-share timber | 0.001 | 0.02 | 0.021 | −0.005 |
| Land-share natural | −0.014 | 0.02 | 0.043 | −0.005 |
| Fisheries production | −0.102 | 0.091 | −0.112 | 0.085 |
| Timber production | 0.003 | 0.01 | 0.018 | 0.009 |
| Manufacturing | −0.031 | 0.029 | −0.026 | 0.034 |

Percentage change in key model variables from two policy scenarios, a one-percent increase in the carbon tax and a two-tier policy consisting of a one-percent increase in the carbon tax together with a one-percent reduction of biofuel subsidies.

*Composite goods.* The expenditure shares of energy services and other inputs are 6.38% and 93.62%, respectively. That is, $\Gamma^Y_{\mathcal{E}_Y} = 0.0638$ and $\Gamma^Y_{M_Y} = 0.9362$.

*Fertilizers.* The expenditure share of energy is 10.95%. The factor share of phosphate is assumed to be a share $\xi_{\mathcal{P}} = 0.5$ out of the factor share of non energy intermediates 62.53%. That is $\Gamma^P_{E_P} = 0.1095$ and $\Gamma^P_{\mathcal{P}} = 0.5 * 0.6253 = 0.3127$. this leaves the expenditure share or other inputs as $\Gamma^P_{M_P} = 0.5778$.

Finally, we need several estimates of elasticities, including the elasticity of substitution, price elasticity of supply and elasticities of conversion costs. For the majority of parameters, we were able to track down estimates from the literature which are presented together with their corresponding reference in Table 4. Where the uncertainty in the estimates were high we employed a wide band for the sensitivity analysis. The parameters that are varied in the sensitivity analysis are indicated as [min, max, and mean] with mean being the baseline values.

**Numerical results**. The full sets of changes in our model quantities and prices resulting from the two policies are presented in Table 5.

We now describe the mapping from changes in model variables to effects on ESPs. For the model variables freshwater ($W$), natural land-use ($L_U$), phosphate ($\mathcal{P}$), and nitrogen (assumed to be proportional to fossil fuel use in fertilizer production $E_P$), there is a simple one-to-one mapping with model variables. For climate change, ocean acidification, biodiversity loss and aerosol loading, however, the mapping is more complicated. For climate change and ocean acidification, we measure the change in pressure on both ESPs as the net change in $CO_2$ emissions. For biosphere integrity, we measure changes in pressure as a change in threats to endangered species (more details on this are given below). We measure aerosol loading as changes in aerosol optical depth. For chemical pollution and ozone depletion, we map pressures to contributing sectors, but do not make any quantitative analysis of the net effects.

Climate change and ocean acidification—are both driven by carbon emissions and we use these emissions as our proxy for the pressures inflicted on these boundaries. To translate changes in model variables into changes in emissions, we use data from refs. [37,38]. From the figure on page 2 of ref. [37] we get the percentage contribution of carbon dioxide emissions per sector outlined in the report. Using these percentages we can thus recover the amount of actual carbon emissions in gigaton carbondioxide (GtCO$_2$) per year connected to a specific variable in our model.

Using this approach, we start by looking at the energy-related emissions that, according to ref. [37], account for a total of 66.5%. Multiplying by the aggregate total emissions in 2005 (44.15 GtCO$_2$) we get 29,36 GtCO$_2$. Next, we allocate these energy-related emissions to the energy service production sector, fossil fuel

extraction, and emissions from fertilizer production. From ref. [37] we have that 6.4% (2.826 GtCO$_2$ eq) of the total energy-related emissions is due to extraction processes. Based on ref. [38], fertilizer production is estimated to cause emissions of 0.575 GtCO$_2$ eq. Hence we can split the total energy-related emission of 29.36 GtCO$_2$ based on these percentages. This implies that 25,960 GtCO$_2$ will be connected to the energy services output in our model, 2.826 GtCO$_2$ is attributed to the fossil fuel extraction process and 0.575 GtCO$_2$ is connected to fertilizer production.

The other emission-related variables in our model are more straightforward. Emissions from industrial processes in ref. [37] are assigned to manufacturing in our model (In total 4.6% = 2.031 GtCO$_2$). Emission from land-use change are assigned to the change in natural land in our model (12.2% = 5.387 GtCO$_2$). Emissions from agriculture are assigned to the total agricultural production variable (13.8% = 6.093 GtCO$_2$). For the fisheries sector,[39] estimate carbon dioxide emissions to be ~0.14 GtCO$_2$.

Using these assignments as a status quo, we can calculate the total policy impact by simply multiplying the percentage change in our model variables resulting from the policy by the status quo emission levels. In total, our model variables cover ~97.4% of the emissions outlined in ref. [37]. The results of this exercise, in terms of percentage changes to each planetary pressure, is outlined in Supplementary Table 2 for the carbon tax policy and Supplementary Table 3 for the combined carbon tax and biofuel tax policy.

To summarize, we find that a 1% increase in the carbon tax leads to a reduction in carbon dioxide emissions by −0.25% or −0.11 GtCO$_2$ yr$^{-1}$, which is what we use as an indicator of the change in pressure accrued to the climate change and ocean acidification boundary. For the combination of carbon and biofuel tax, the change is −0.26% or −0.12 GtCO$_2$ yr$^{-1}$.

Biodiversity loss—is a notoriously difficult task to assess at a global scale. Studies that quantify terrestrial biodiversity losses resulting from the environmental pressures of human activities typically focus on land-related impacts[40,41]. There are, however, multiple other environmental pressures causing loss of biodiversity that are not related to land-use[42]. In ref. [3] the global extinction rate is used as one way of quantifying this boundary (defined as extinctions per million species-years). Here, we will make use of the IUCN Red List of Threatened Species to derive a measure of biodiversity loss. The Red List identifies not only the species that have been confirmed to have gone extinct but also the species that are currently

threatened and, if pressures remain, may become extinct in the future.[43] identify the drivers behind the prevalent threats to the species on the Red List in a comprehensive assessment of more than 8000 species. These drivers can be directly identified as variables in our model. In ref. [43] there is overlap between threats in the sense that multiple activities can pose threats to a given species. We refer to a decrease in an activity posing a threat to a certain number of species as a decrease in threats. Without knowing the overlap between threats, we can not translate this into changes in number of threatened species. Therefore, we use the change in threats as our measure. Agricultural activity poses threats to 5295 species, which is the largest number of threats. The second-largest threat comes from logging, which threatens 4049 species, and we assign this to timber production in our model. Apart from those, we make the following assignments. Pollution from agriculture threatens 1523 of the species and this is assigned to fertilizer production. Over exploitation (fishing), threatening 1118 species, is assigned to fisheries production. Energy production (oil and gas) and renewable energy production account for threats to 56 species, which we assign to fossil fuel extraction and renewables. Finally, threats from urban development (industrial), pollution (except agriculture), human disturbance (work), transport, energy production (mining) summed to 3573 which we assign to manufacturing. There are also significant biodiversity effects of climate change, which threatens 1688 of the species. In this analysis, we abstract from the effects of changes in one ESP on other ESPs (unless the ESP is directly captured by a model variable). We can note, however, that including the effects of climate change would lead to larger decreases of biodiversity loss.

Hence, having connected the categories of threats to species by driver in ref. [43] to our model variables, we can measure the biodiversity impact of a policy by assessing whether the number of threats increases or declines as a result of the policy. For example, if the agricultural production increases by 1% as a result of a policy in our model then this would increase the number of threats from agricultural activity by 52.95 ($0.01 \times 5296$).

The results, in terms of percentage change to the number of threatened species, are outlined in the column labeled Biodiv. in Supplementary Table 2 for the carbon tax policy and Supplementary Table 3 for the combined carbon tax and biofuel subsidy removal. To summarize, this implies that the total number of threats decline by 0.018% for the carbon tax and by 0.011% for the combined carbon and biofuel tax.

Finally, it should be noted that there are indeed several caveats to our approach for assessing biodiversity loss. First, it should be noted that this measure of biodiversity loss is just a proxy for true biodiversity loss. Future work would benefit from assessing the drivers of the actual rate of species loss as defined in e.g., ref. [1]. Furthermore, we have taken the description of threats in[43] and mapped them to our model variables. For instance, all threats assigned to agriculture in ref. [43] are assigned to agricultural production in our model. Perhaps some part of these threats come from land use change associated with agriculture rather than agriculture as such. In that case they should be mapped to our land use variables. We do not have a proper basis for such reassignment and, therefore, stick close to their assignment. Qualitatively, this distinction could matter for the carbon tax in isolation, but will not be important for the carbon tax combined with biofuel policy.

Aerosol loading—is proxied following[3] which use aerosol optical depth (AOD) as an indicative measure of planetary pressure. To determine how AOD changes as a result of policy, we use data from three sources[44–46]. The impact is calculated as follows. First, we calculate a global average estimate of AOD from the main regional anthropogenic sources (sulfur (0.0392), black carbon (0.0003) and organic carbon (0.0011)) provided by ref. [44]. Second, we use data from ref. [45] to calculate the share of global aerosol contributing emissions for each of these respective sources (sulfur (3.6%), black carbon (32%) and organic carbon (63%)) that stem specifically from biomass burning (assuming that approximately 90% of biomass burning emissions result from land-use change[46]). Third, using these estimates, we can calculate the amount of global AOD which ought to be attributed to emissions from fossil fuel and biofuels (0.038) and biomass burning (0.0022). These estimates are then connected to the model variables fossil fuel consumption (in energy services, fertilizer production and fisheries), biofuel production and change in natural land. In total, a 1% carbon tax leads to a 0.0136% ($-5.5 \times 10^{-6}$) decline in AOD and the combined carbon tax and biofuel policy leads to a decline of 0.014% ($-5.7 \times 10^{-6}$) (further details can be found in Supplementary Tables 2 and Table 3).

Stratospheric ozone depletion and chemical pollution—are not directly quantified in terms of their effects on the boundaries. Stratospheric ozone depletion increases with $N_2O$ emissions from agricultural production, fossil fuel use, manufacturing, and biofuels. For an increase in the carbon tax all these activities except for biofuel use decreases. The net effect is thus potentially ambiguous. If the carbon tax increase is complemented with a decrease in biofuel subsidies, all relevant variables decrease and we conclude that the net effect is a decrease in the pressure.

*Chemical pollution.* Chemical pollution increases in manufacturing, extracted fossil fuels, total agricultural production, agricultural production for food, fossil fuel use in fertilizer production, and fossil fuel use in energy services production. All these activities decrease with a carbon tax, with or without a biofuel policy. Hence, we conclude that chemical pollution will decrease in both cases.

For the remaining boundaries—the impacts are easier to assess since they are directly tied to specific model variables. First, the impact on the biogeochemical flows is assigned to the model variables phosphate and fossil-fuel use in fertilizer production. While the former is self-explanatory, the latter is used as a proxy for

nitrogen, which relies almost entirely on fossil fuels in its production. For phosphorus, we translate the change into Gg P yr$^{-1}$ using the value for current flows (mined and applied to erodible soils) from ref. [3]: $-0.000068 \times 14,000 \approx -0.9$ Gg P yr$^{-1}$ for the carbon tax and $-0.0005 \times 14,000 \approx -7$ Gg P yr$^{-1}$ for the combined carbon and biofuel policy. For nitrogen, we translate the change into TgN yr$^{-1}$ using ref. [3]: $-0.0013 \times 150 \approx -0.2$ TgN yr$^{-1}$ for the carbon tax and $-0.0018 \times 150 \approx -0.28$ TgN yr$^{-1}$ for the combined carbon and biofuel policy. Second, for the land-system boundary, we rely on the model variable natural land use as an indicator of the direction this boundary is moving in. This is translated into MHa using the average (between high and low value) for "natural forests" in ref. [47]: $-0.00014 \times 3507 \approx -0.5$ MHa for the carbon tax and $0.00043 \times 3507 \approx$ 1.5 MHa for the combined carbon and biofuel policy. Third, the freshwater boundary is directly tied to the water variable in our model. We translate this into km$^3$ yr$^{-1}$ using the value for current use in ref. [3], we the reduction is given by $2600 \times 0.00009 \approx 0.24$ km$^3$ yr$^{-1}$ for the carbon tax policy and $-2600 \times 0.00036 \approx -0.93$ km$^3$ yr$^{-1}$ combined carbon tax and biofuel policy.

**Reporting summary**. Further information on research design is available in the Nature Research Reporting Summary linked to this article.

## Data availability

All model parameters and data needed to replicate the results in this article are stated explicitly in the "Methods" and Supplementary Methods.

## Code availability

Model code written in Python is available for download at https://github.com/engstromgustav/carbonpricing_and_planetaryboundaries.

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

## Acknowledgements

We are grateful for comments received from seminar participants at WCERE, SLU, CER-ETH, CEPE, Uppsala University, Aalto University, National University of Singapore, RATIO, University of Oslo, University of Bergen, IHP workshop at the Henri Poincaré instutute in Paris, and the Beijer Institute. We are also thankful to detailed comments by two anonymous reviewers. Engström, Gars and Kiran also acknowledges funding support from the Ragnar Söderberg Foundation (E50/14).

## Author contributions

The idea was conceived by G.E., J.G., C.K., D.S., R.C., and T.L. The formal analysis was done by J.G., G.E., C.K., and D.S. Code and numerical computations was done by G.E. and J.G. The paper was written by G.E., J.G., C.K., D.S., R.C., and T.L. Data curated from GTAP by B.N.

## Competing interests

The authors declare no competing interests.
