## [Peer Review File · Nature Communications]

REVIEWER COMMENTS

Reviewer #1 (Remarks to the Author):

I like your viewpoint on your paper. However, several issues should be responded.

1.It is not clear to me why you have jumped on the biofuel from the carbon tax? You have mentioned that "...higher fossil fuel prices make biofuel production more competitive."; however, is it enough for focusing on biofuel? If you remove biofuel and add other RE what will be happened in the key message of your paper! I am concern about it.

2.Same as the above comment, the title is "A carbon tax with planetary boundaries," but we can see a cristal role of biofuel in the article. Where is the effect of biofuel on your title? Title should be modified or biofuel should be...

3.When you talk about the carbon tax, you are talking about a causal for at least five of the nine boundary processes. How can you be sure that the carbon tax is a suitable policy option for consideration? Can I replace it with tarrifs for example? This should be strongly clarified. If you see the policies of the governments, the carbon tax is not in the priority due to the negative effects that it may have on the industries!

4.You have used GTAP, which can be reasonable choice. However, you have not figured out which parameters drive the results? Indeed, you need to focus exclusively on that parameter.

5.There are many parameters in the GTAP model, and they are all uncertain or you have not considered it! You should explain it to them. The issues in the sensitivity analysis are not sufficient.

6.How can you trust the GTAP data? Year of the data? Your analysis is highly dependent on those data.

Comments on “NCOMMS-20-00560-T A Carbon Tax with Planetary Boundaries”

Summary

This manuscript asks whether a small increase in global carbon prices would cause market responses that affect other environmental goods and services than the climate. The environmental goods and services considered here are the nine earth system processes (ESPs) whose failure would risk catastrophic outcomes for humankind¹.

The authors develop a partial model of the world economy, including the ten sectors that contribute most to the degradation of these ESPs. Each sector is modeled as a representative firm optimizing production for given out- and input prices. A representative consumer maximizes utility by purchasing food (agriculture and fish), manufactured goods, recreational land and timber. Sectors interact by competing in output- and input markets.

The immediate impact of the higher carbon price is to lower the producer price of fossil fuels, *ceteris paribus* making fossil fuels more expensive for the three sectors that buy them. ESPs are affected by adjusted production decisions in the various sectors of the economy.

The results are derived analytically by comparative statics. The authors quantify the parameters using various sources, primarily the GTAP database. Changes in the use of freshwater, land, phosphorus and nitrogen translate directly into the corresponding ESPs. For the other ESPs, the authors calculate how the adjustments in activity in each sector translate into changes in emissions or environmental pressures using different data sources.

The main finding is that an increase of a global carbon price from zero by one percentage point of the fossil fuel price decreases pressure on all but two ESPs by 0.01% to .25%. For freshwater use and land-system change, pressure increases by 0.01% and 0.006% respectively. If in addition subsidies to biofuels were reduced by one percentage point, pressure on all ESPs is reduced.

Assessment

In this accessibly written manuscript, the authors address an important economic question. We know that climate policy has potentially large co-benefits from reducing air pollution², and we are aware of concerns that climate policy-induced demand for biofuels could lead to undesirable land conversion³. The authors build a model economy to investigate which of all the ESPs a carbon price could impact through market adjustments.

To make their case for the need for such a model even stronger, I would appreciate if the authors could argue more explicitly why we a priori expect economically significant effects. From an economic theory perspective, the theory of second best⁴ comes to mind: Correcting for one market imperfection may have negative welfare impacts in an economy in which other distortions are also present. From a policy perspective, mentioning more

¹ <https://www.nature.com/articles/461472a>

² <https://www.nature.com/articles/nclimate2009>

³ <https://onlinelibrary.wiley.com/doi/full/10.1111/sjoe.12177>

⁴ <https://www.jstor.org/stable/pdf/2296233.pdf>

examples (like palm oil production threatening biodiversity⁵) may help the reader understand the need for a joint economic assessment of these environmental problems.

Any analytic economic model needs strong assumptions; decisions about what to include, the level of detail, etc. It is hard to argue that any specific model is “the most useful” for a question at hand. The authors make some clear and well-argued choices.

First, the model is global, hence so is the carbon price. Currently only around 20% of CO₂ emissions globally are covered by carbon pricing, with prices ranging from US\$ 1 to US\$ 127⁶. A global carbon price is not in sight. To consider this hypothetical policy is still meaningful in the sense that any more complicated model would probably not add much insight for the modeling cost incurred. But it raises questions as to how to interpret the numbers generated, as they do not correspond to any real-world policy option. I generally read the results as indicating which ESPs one needs to be more or less concerned about when (if) climate policy becomes more ambitious.

Second, the authors look at a marginal carbon price increase in a static model with a time frame of 5-10 years. They explicitly limit their analysis to the short-term substitution patterns arising from re-optimizing input demand- and supply decisions for given technologies: substituting away from fossil fuels and producing a little less overall. Those responses point at the ESPs that might be immediately affected. Any substantial increase in carbon prices would (one hopes) lead to technological and industrial changes, so that the results would not scale.

Third, the model is by necessity highly aggregate. For some of the local issues (freshwater scarcity and land degradation?) there may be a lot of heterogeneity and the policy change may even have opposing signs locally. This is not an argument against an aggregate model, but I would appreciate the authors’ thoughts.

Fourth, are the relations modeled the most relevant ones? The authors choose the sectors that “account for more than 90% of the drivers for the majority of the planetary pressures” (line 96). That seems like a good criterion to me. From an economic perspective the absence of a “service sector” seems a strong choice – wouldn’t it in principle be possible that demand and hence factors are re-allocated to such a sector in response to carbon pricing, potentially easing pressures on the environment? I would appreciate if the authors could address the absence of the sector explicitly.

Overall, I think it would be possible to make reasonable, alternative assumptions that would yield quantitatively or even qualitatively different results. For example, one could argue that the carbon price increase foreshadows more stringent future climate policies such that fossil fuel supply does not change or even increases (the green paradox⁷). That is not a limitation of the current paper, but rather a feature of any economic modeling of this kind. For example, the authors refer to the highly influential integrated assessment literature, in which reasonable adjustments in parameters result in the carbon price changing by orders of magnitude. I think the authors here make careful, well-argued choices.

⁵ <https://www.iucn.org/resources/issues-briefs/palm-oil-and-biodiversity>

⁶ <https://openknowledge.worldbank.org/bitstream/handle/10986/31755/211435KeyFigures.pdf>

⁷ <https://academic.oup.com/reep/article/9/2/246/1626618>

Specific comments:

- Title: I would suggest using carbon *price* rather than *tax* as it is more general, encompassing cap-and-trade schemes.
- Line 23: The ESPs should be named explicitly from the start, earlier than on page 3.
- Line 24: What is a “catastrophic consequence” in economic terms in this context?
- Line 26: The “the complexity of the many interlocking processes”: In this paper, no interactions between the ESPs are included?
- Line 49: It might strengthen your argument to explicitly address why you do not want to use a version of the GTAP (or another) general equilibrium model to derive the economic impacts of a carbon price but rather prefer to only use the GTAP data in a new, partial model.
- Line 50: Please explain what “transparent enough to enable independent assessment of the validity of the results” means in this context. Specifically, how do you define “valid results” here?
- Line 54: “most ESPs” – please spell out the number exactly.
- Line 69: “merely 4% of global economic output” undersells the importance of food (the paradox of value).
- Line 82-85: “The design is the conscious result of...” I would argue that is the aim of any analytic economic model and hence the statement maybe is not needed? Or could you be more specific?
- Line 88-91: I think it would be more accurate to write that DICE and IMAGE are models trying to answer different research questions and hence necessarily have a rather different structure.
- Line 96: “account for more than 90% of the drivers for the *majority* of the planetary pressures”. The statement is a little ambiguous. Could you be more specific?
- Line 108: In what way are the parameters easier to interpret in this analysis?
- Line 110: Wouldn’t it help the reader if you added a simplified example of how you combine the numerical parameters to get the policy impact for a specific ESP?
- Line 111-112: Again, IAMs answer a different question. Hence the current model cannot be easily compared to them according to these criteria. There are simplified analytic IAMs that allow a similar level of intuitive insight⁸.
- Line 129: In which sense are the parameter values “coherent” because they stem *mostly* from the same source?
- Line 134: You assume counterfactually that the current carbon price in your model is zero, yet there are carbon prices currently in place. How does that affect your interpretation of your results?
- Line 142: It would be nice if you could compare your elasticity of CO₂ emissions with respect to the carbon price to other existing estimates for specific sectors or countries to see if your model produces results that are similar to empirical estimates⁹
- Figure 2: Please explain the colors in the figure.
- Line 162: “...a combination of a number of different effects”. Under “Drivers of planetary pressures”, the SI only mentions the same ones as the text? The SI does

⁸ https://papers.ssrn.com/sol3/papers.cfm?abstract_id=3374584

⁹ for example <https://www.aeaweb.org/research/carbon-tax-impact-sweden>

not feature a section “Determining how policy-induced changes impact the planetary pressures”.

- Line 176: “...increased freshwater use...” I consider the freshwater sector a good case for which the authors could explain the impact of their simplifying assumptions intuitively. For example, not all farms around the world have access to irrigation and hence cannot substitute fossil fuel for freshwater¹⁰.
- Line 178-9: I think it would be good to point out the relative magnitude of the effects for all ESPs that are not directly impacted but via linkages between economic sectors.
- Line 204-5: “Other types of biofuels, that are produced differently, would perhaps fit better into our renewables-category”. Could you state an estimate of the share of biofuels currently on the market that are competing with food for land? Also, it might be worth pointing out that for example the EU Renewable Energy Directive states the type of land that may be converted to biofuel production¹¹, potentially already circumventing the problem identified here?
- Line 220: The table reference is broken (as are others in the SI).
- Line 243-244 “Our analysis clearly shows that indeed these interacting effects can be quite substantial and should not be ignored”. Where the effects are not “direct” (as they are for ocean acidification and climate change, -0.25%), the effects are at least one order of magnitude smaller (biodiversity being the largest with 0.027%). The exception are nitrogen flows (-0.14%, where link is “rather direct” as natural gas is its most important input by far).
- Line 363: “About 50”. Please state the exact number.

¹⁰ <https://pubs.aeaweb.org/doi/pdf/10.1257/0002828053828455>

¹¹ https://ec.europa.eu/energy/topics/renewable-energy/biofuels/sustainability-criteria_en

Reviewer #1

General comments: We thank the reviewer for many valuable suggestions, in particular for raising the question of the relationship of our model to the GTAP database, and the role played by biofuels in our model. A point-by-point reply to specific questions raised are provided below. Note that the review process also led us to update two of our parameter estimates, i) the elasticity of substitution between various energy sources (source: Papageorgiou et.al., 2017)), and ii) the energy share in agriculture (source: FAO Agricultural Outlook report 2016-2025). These updates only had minor effects on some of our outcome variables. They did however, not, change any qualitative results, implying that all previous conclusions still hold. The new parameter values can be found in table 5 (σ_ϵ) and 3 (Q_ϵ, ϵ_a) in the methods section.

I like your viewpoint on your paper. However, several issues should be responded.

1. It is not clear to me why you have jumped on the biofuel from the carbon tax? You have mentioned that "...higher fossil fuel prices make biofuel production more competitive."; however, is it enough for focusing on biofuel? If you remove biofuel and add other RE what will be happened in the key message of your paper! I am concern about it.

Reply: We agree that the reason for using a biofuel policy as a complementary (we call it "auxiliary" in the manuscript) policy to carbon pricing needs a good explanation. Our model does indeed include other renewable energy sources (see eq. (8), (2), Methods section). In the revised manuscript, we indicate this more explicitly (p.6 line 99-100, "renewable energy (other than biofuels)"), and again in p.19, Methods (line 350, just prior to eq. (8)). Briefly, as the reviewer surmises, the use of renewables increases significantly in response to the carbon tax (an increase of 0.466%, see row "Renewables (Other than biofuel)" in Table 6, p.42). Likewise, biofuel use also increases (by 0.71%, same table) to compensate for the higher relative price of fossil fuels. The decrease in fossil fuel use is thus compensated for by increasing production of both biofuels and renewables (note: agricultural output also adjusts).

The primary motivation for our focus on biofuels, as opposed to other renewables, is that biofuel alone is a direct agricultural product, meaning that there is a unique link to agriculture that is not present for other renewable energy sources. As a result, an increase in biofuel production (as a result of the carbon tax) has significant effects on other planetary pressures, in particular land use. In contrast, the supply of other forms of renewable energy has no direct effects (to the best of our knowledge) on other planetary pressures (i.e. there is only a benefit, from the "climate effect" resulting from the replacement of fossil fuels). The introduction of a biofuel policy is thus motivated on the basis that it attenuates the primary mechanism by which a carbon tax could conceivably increase planetary pressures.

We highlight our reasons for focusing on biofuels and biofuel policy more explicitly in the revised manuscript (abstract, line 14-17; and Main, p.4, line 62-66).

2. Same as the above comment, the title is “A carbon tax with planetary boundaries,” but we can see a crystal role of biofuel in the article. Where is the effect of biofuel on your title? Title should be modified or biofuel should be...

Reply: We agree that the choice of title is important and should reflect the article’s central message. We have considered various options and decided to use the somewhat more general title “Carbon pricing and Planetary Boundaries”. We wanted a short and clear title. The focus of the analysis is the consequences for the planetary boundaries of carbon pricing. We see the biofuel policy as a complementary policy that we consider based on the found consequences of a carbon tax in isolation. We did not manage to find a succinct title that included biofuels.

3. When you talk about the carbon tax, you are talking about a causal for at least five of the nine boundary processes. How can you be sure that the carbon tax is a suitable policy option for consideration? Can I replace it with tariffs for example? This should be strongly clarified. If you see the policies of the governments, the carbon tax is not in the priority due to the negative effects that it may have on the industries!

Reply: Given the centrality of the carbon tax for our analysis, we agree that it is indeed important to be clear about what our analysis actually covers. There seems to be two issues to address here. The first is our reference to a carbon tax specifically, as opposed to other instruments for pricing carbon. We initially used the terminology of a “carbon tax” mainly because this mirrors how the price is implemented in our model, but the revised manuscript now refers to a “carbon price” wherever appropriate. On p. 8, line 145-147, of the revised manuscript, we explicitly state that our analysis with a carbon price is equivalent to one considering any other policy that effectively puts a price on carbon emissions (e.g. a cap-and-trade system).

The second issue is whether it is appropriate to focus on carbon pricing as opposed to some other policy instruments. There are at least three reasons for this. First, while there is currently no global carbon tax and one is unlikely in the near future, it is probably the most widespread and salient climate change policy. According to the most recent estimates, nearly a quarter of global greenhouse gas emissions are currently covered by some form of carbon pricing.. Second, a carbon tax has for quite some time been the policy preferred by many scholars as the most efficient policy option for combating global warming. It is also the policy option most commonly analyzed in integrated assessment models of climate economy interaction. Focusing on a carbon tax thus makes our approach and results more comparable to other modelling exercises. Finally, in any model as aggregated as ours, the details of the policy instrument are somewhat less important. Any policy that has the effect of making carbon emissions more expensive, whether or not by imposing a price on carbon explicitly, is more or less equivalent at this level of aggregation. Hence our analysis also applies to many alternative policies.

4. You have used GTAP, which can be reasonable choice. However, you have not figured out which parameters drive the results? Indeed, you need to focus exclusively on that parameter.

Reply: Thanks! This was a useful suggestion. To figure out which parameter drives the result we took the following approach. We iterated over all parameters of our model, and for each iteration we either increased or decreased each parameter estimate by a fixed percentage (described in SI p.20, from line 304). We then ran a simulation using this perturbed parameter estimate and compared the model results to our baseline results by calculating the mean relative change of the policy responsiveness of our model variables as a result of the increase or decrease in the specific parameter estimate. Finally, we ranked all the parameters based on the largest mean relative change of the policy responsiveness of our model variables. The results of this analysis indicates that the model results are most sensitive to the elasticity of substitution parameters and the supply elasticity of fossil fuels. This led us to include two new parameters (elasticity of substitution in energy production and supply elasticity of energy) in our final sensitivity analysis. The reason we had previously excluded these variables was that we felt that these parameters were fairly well pinned down in the literature. Overall, this extended sensitivity analysis did not change overall conclusions from our previous sensitivity analysis. We have added an explanation of the approach taken to find the most central parameters in the sensitivity analysis section of the SI.

5. There are many parameters in the GTAP model, and they are all uncertain or you have not considered it! You should explain it to them. The issues in the sensitivity analysis are not sufficient.

Reply: The reviewer raises an interesting point, regarding parameter uncertainty in the GTAP. Before we address this point, we note that our analysis largely uses *data* (not parameters) from the GTAP database. In particular, aggregate sectoral cost and quantity shares, which, as pointed out below (see our response to q.6 immediately below), is widely used for its reliability and consistency. We only use two parameters from the GTAP model, the elasticities of substitution in timber production and fisheries. These two parameters, which are chosen with rather generous intervals, are part of parameters whose uncertainty we account for in our sensitivity analysis (see table 5 in revised manuscript).

These aspects are explicitly mentioned in the revised manuscript (p.8, lines 135-138). Further details on how we use the GTAP database are discussed in the “Expenditure Shares” subsection (p.23-24) of “Data and Parametrization” section in Methods.

6. How can you trust the GTAP data? Year of the data? Your analysis is highly dependent on those data.

Reply: The reviewer is perfectly correct in pointing out the centrality of GTAP data to our analysis. Our analysis uses data from the GTAP database for the year 2014.

As to its reliability, the GTAP database is one of the most peer-reviewed, cross-checked and verified global economic datasets available publicly; every round of it is carefully reviewed for 3-4 years by academics, policy-makers and industry experts before being released to the public (Aguiar et al 2020; Aguiar, Narayanan and McDougall, 2016; Walmsley et al 2018). It is widely considered to be the most consistent global input output dataset, and is the most commonly used database across a wide variety of studies, including by institutions such as the European Commission¹, the US Department of Commerce², the World Bank (e.g. Mani et al 2018³), etc., by leading consulting firms such as McKinsey⁴, PWC⁵, KPMG⁶, etc., and in papers published in leading academic journals (Hertel, 2016; Moore et al 2017; Nelson et al 2014; Elliott et al 2010, etc.) covering subjects as diverse as agricultural trade, climate change, newer technologies, energy markets and water. While no database is perfect, GTAP is among the best resources available and its limitations are well understood. Using it as the basis for our study also makes our results more comparable to, and more accessible to, the broad range of scholars and institutions that trust and have worked extensively with these data.

References:

AGUIAR, Angel et al. The GTAP Data Base: Version 10. **Journal of Global Economic Analysis**, [S.I.], v. 4, n. 1, p. 1-27, June 2019. ISSN 2377-2999. Available at: <<https://www.gtap.agecon.purdue.edu/resources/jgea/ojs/index.php/jgea/article/view/77>>. Date accessed: 04 May 2020. doi:<http://dx.doi.org/10.21642/JGEA.040101AF>.

Aguiar, Angel, Badri Narayanan, & Robert McDougall. "An Overview of the GTAP 9 Data Base." **Journal of Global Economic Analysis** 1, no. 1 (June 3, 2016): 181-208. Available at: <<https://jgea.org/resources/jgea/ojs/index.php/jgea/article/view/23>>. Date accessed: 04 May 2020. doi:<http://dx.doi.org/10.21642/JGEA.010103AF>

Elliott et al 2010 <https://pubs.aeaweb.org/doi/pdf/10.1257%2Faer.100.2.465>

¹ See e.g.

<https://ec.europa.eu/jrc/en/event/workshop/global-agricultural-domestic-support-economic-modelling>).

² See <https://www.usitc.gov/publications/332/pub4703.pdf>.

³<http://documents.worldbank.org/curated/en/705871522683196873/Paris-climate-agreement-and-the-global-economy-winners-and-losers>

⁴<https://www.mckinsey.com/~media/mckinsey/featured%20insights/future%20of%20organizations/australias%20automation%20opportunity%20reigniting%20productivity%20and%20inclusive%20income%20growth/australia-automation-opportunity-vf.ashx>

⁵<https://www.pwc.com/gx/en/issues/data-and-analytics/publications/artificial-intelligence-study/research-and-methodology.html>

⁶ https://www.gtap.agecon.purdue.edu/resources/res_display.asp?RecordID=5249

Hertel, T. Food security under climate change. *Nature Clim Change* 6, 10–13 (2016).

<https://doi.org/10.1038/nclimate2834>

<https://www.nature.com/articles/nclimate2834?proof=true&platform=oscar&draft=collection#Bib1>

Moore, F.C., Baldos, U., Hertel, T. *et al.* New science of climate change impacts on agriculture implies higher social cost of carbon. *Nat Commun* 8, 1607 (2017).

<https://doi.org/10.1038/s41467-017-01792-x>

Nelson et al 2014: <https://www.pnas.org/content/111/9/3274>

Terrie Walmsley, Badri Narayanan, Angel Aguiar & Robert McDougall (2018) Building a global database: consequences for the national I–O data, *Economic Systems Research*, 30:4, 478-496, DOI: [10.1080/09535314.2018.1440533](https://doi.org/10.1080/09535314.2018.1440533)

Reviewer #2

General comments: We thank the reviewer for the very careful reading of our draft manuscript and for providing very constructive suggestions. We provide a point-by-point reply below. We note that the review process also led us to update two of our parameter estimates: i) the elasticity of substitution between various energy sources (source: Papageorgiou et.al., 2017)); and ii) the energy share in agriculture (source: FAO Agricultural Outlook report 2016-2025). These updates only had minor effects on some of our outcome variables. They did however, not, change any of our qualitative results, implying that all previous conclusions still hold. The new parameter values can be found in tables 5 (σ_ε) and 3 ($Q_\varepsilon, \varepsilon_a$) in the Methods section.

Comments on “NCOMMS-20-00560-T A Carbon Tax with Planetary Boundaries”

Summary

This manuscript asks whether a small increase in global carbon prices would cause market responses that affect other environmental goods and services than the climate. The environmental goods and services considered here are the nine earth system processes (ESPs) whose failure would risk catastrophic outcomes for humankind .

The authors develop a partial model of the world economy, including the ten sectors that contribute most to the degradation of these ESPs. Each sector is modeled as a representative firm optimizing production for given out- and input prices. A representative consumer maximizes utility by purchasing food (agriculture and fish), manufactured goods, recreational land and timber. Sectors interact by competing in output- and input markets.

The immediate impact of the higher carbon price is to lower the producer price of fossil fuels, ceteris paribus making fossil fuels more expensive for the three sectors that buy them. ESPs are affected by adjusted production decisions in the various sectors of the economy.

The results are derived analytically by comparative statics. The authors quantify the parameters using various sources, primarily the GTAP database.

Changes in the use of freshwater, land, phosphorus and nitrogen translate directly into the corresponding ESPs. For the other ESPs, the authors calculate how the adjustments in activity in each sector translate into changes in emissions or environmental pressures using different data sources.

The main finding is that an increase of a global carbon price from zero by one percentage point of the fossil fuel price decreases pressure on all but two ESPs by 0.01% to .25%. For freshwater

use and land-system change, pressure increases by 0.01% and 0.006% respectively. If in addition subsidies to biofuels were reduced by one percentage point, pressure on all ESPs is reduced.

Assessment

In this accessibly written manuscript, the authors address an important economic question. We know that climate policy has potentially large co-benefits from reducing air pollution, and we are aware of concerns that climate policy-induced demand for biofuels could lead to undesirable land conversion. The authors build a model economy to investigate which of all the ESPs a carbon price could impact through market adjustments.

- 1. To make their case for the need for such a model even stronger, I would appreciate if the authors could argue more explicitly why we a priori expect economically significant effects. From an economic theory perspective, the theory of second best comes to mind: Correcting for one market imperfection may have negative welfare impacts in an economy in which other distortions are also present. From a policy perspective, mentioning more examples (like palm oil production threatening biodiversity) may help the reader understand the need for a joint economic assessment of these environmental problems.*

Reply: We thank the reviewer for raising an interesting conceptual point, and for providing a great example as well. We have now added the palm oil/biodiversity trade-off to help better illustrate the challenges involved in making policies pertaining to one earth system process without necessarily affecting another (see p.2, line 34-36, of revised manuscript).

As for the underlying question, we would submit that the strength of the different economic effects are not a priori obvious: while a few effects are anticipated to be large (e.g. the effect of carbon tax upon fossil fuels), some others to be small but significant (e.g. effect of carbon tax on fertilizer), many others depend upon interactions and substitution parameters to such a degree that their magnitudes are difficult to anticipate (e.g. how the land use in agriculture decision responds to climate policies). Moreover, some of these effects have not been thoroughly explored before, meaning there is little basis for forming expectations regarding the strength of the effects. Yet, despite the lack of information regarding the magnitude of different economic effects, the planetary boundaries literature advocates stringent policies to prevent boundary crossings (or to bring processes back to within the boundary, should it have been already crossed). Our investigation is centered around exploring which economic effects are more important, and what relationships matter most, in the hopes that we might contribute some sense of their magnitudes to the current debate. We have edited paragraph 2 on p.3 (along with footnote 4) with this in mind. This, along with the slightly edited text in the second paragraph of

“Model Description” (p.5-6) should clarify that, on the whole, many of the relevant linkages have not been investigated and therefore little is therefore known regarding the magnitude of effects.

Finally, the theory of the second best certainly bears a close relation to our investigations. As the reviewer points out, correcting one (of many) market imperfections need not be welfare enhancing, and this is very likely the case in our setting. In our case, however, performing welfare analysis would require us to not just parameterise the welfare linkages to the Earth System Processes themselves, but also the strength and interaction of the economic externalities they impose. In view of the difficulty in parameterising even a single damage function (e.g. the effect of higher temperatures on economic output), providing such an analysis would be a heroic undertaking at the present moment. We leave the task of performing a full welfare analysis and finding optimal policies when faced with the multiple externalities exemplified by the planetary boundaries to a later study. Our focus here was on understanding the degree to which addressing the strongest of these externalities, climate change, with the simplest and most widely used benchmark tool, carbon taxes, affects the other planetary boundaries, where we do not explicitly take account of the optimality of the policy measure (increased carbon taxes or reduced biofuel subsidies). Clearly, the last word on this subject has not been written.

Any analytic economic model needs strong assumptions; decisions about what to include, the level of detail, etc. It is hard to argue that any specific model is “the most useful” for a question at hand. The authors make some clear and well-argued choices.

- 2. First, the model is global, hence so is the carbon price. Currently only around 20% of CO₂ emissions globally are covered by carbon pricing, with prices ranging from US\$ 1 to US\$ 127 . A global carbon price is not in sight. To consider this hypothetical policy is still meaningful in the sense that any more complicated model would probably not add much insight for the modeling cost incurred. But it raises questions as to how to interpret the numbers generated, as they do not correspond to any real-world policy option. I generally read the results as indicating which ESPs one needs to be more or less concerned about when (if) climate policy becomes more ambitious.*

Reply: The reviewer’s comments are consistent with our own thoughts. The results should indeed be interpreted as indicating which ESPs one needs to be more or less concerned about when (if) climate policy becomes more ambitious. We have tried to clarify this in footnote 3 in the introduction.

- 3. Second, the authors look at a marginal carbon price increase in a static model with a time frame of 5-10 years. They explicitly limit their analysis to the short-term substitution patterns arising from re-optimizing input demand- and supply decisions for given technologies: substituting away from fossil fuels and producing a little less overall. Those responses point at the ESPs that might be immediately affected. Any substantial*

increase in carbon prices would (one hopes) lead to technological and industrial changes, so that the results would not scale.

Reply: This is absolutely correct. Our perspective is that a carbon price will always form a benchmark (implicit or explicit) for other (possibly second-best) policies. In the absence of any single global policy, and with a mix of region-specific policies (tax/subsidy/cap-and-trade), it is not evident that any other benchmark policy is either available or likely to yield added insights. The second point is also perfectly valid: addressing the full long term consequences of the policies would require modeling (at least) both endogenous technical change and the previously mentioned strength of externalities. The development of a dynamic multi-sector model incorporating both aspects is a key research goal for the future.

4. *Third, the model is by necessity highly aggregate. For some of the local issues (freshwater scarcity and land degradation) there may be a lot of heterogeneity and the policy change may even have opposing signs locally. This is not an argument against an aggregate model, but I would appreciate the authors' thoughts.*

Reply: We are thankful to the reviewer for raising this point. The two ESPs noted by the reviewer, freshwater and land use, are indeed rather regional in nature. As noted in the original planetary boundary studies, freshwater use at a global scale is not particularly unbalanced, and is only a challenge regionally, often even sub-regionally, in the semi-arid regions that are the most populated today (India, China, Sub-saharan Africa, semi-arid parts of the Western U.S and Australia, South Africa) and often temporally. Similarly, the adverse land use changes our analysis finds and discusses are in fact largely limited to tropical and sub-tropical regions (with many studies reporting a reconversion from agricultural to natural land outside the tropics, Song et al (2018)). In fact, it may be more appropriate for many regions to talk of the net environmental load of agricultural activities as being a key driver of policy, in view of the local environmental implications being far larger than the global. To illustrate, for certain regions, key environmental indicators (nutrient loading, availability of sufficient arable land) may either suggest either a reduction or at least a rationalisation of all chemical input and resource usage (North western India, Western/Eastern U.S., China, see e.g. Chand and Pavitra (2015), Vitousek et al. (2009)) while for many parts of sub-saharan Africa, the magnitude of the yield gaps may suggest an increased usage of certain inputs, in particular fertilizers and other chemical inputs (McArthur and Mccord (2017)). Clearly, therefore, policy even changes direction regionally, as the reviewer alludes to.

In view of the global nature of our model framework, and paucity of space, this issue was not highlighted in the previous draft of the manuscript. Should the reviewer and editor feel that this discussion is very relevant, we can of course incorporate it into the manuscript.

Song, X. P., Hansen, M. C., Stehman, S. V., Potapov, P. V., Tyukavina, A., Vermote, E. F., & Townshend, J. R. (2018). Global land change from 1982 to 2016. *Nature*, 560(7720), 639-643.

McArthur, J. W., & McCord, G. C. (2017). Fertilizing growth: Agricultural inputs and their effects in economic development. *Journal of development economics*, 127, 133-152.

Chand, R., & Pavithra, S. (2015). Fertiliser use and imbalance in India. *Economic & Political Weekly*, 50(44), 99.

Vitousek, P. M., Naylor, R., Crews, T., David, M. B., Drinkwater, L. E., Holland, E., ... & Nziguheba, G. (2009). Nutrient imbalances in agricultural development. *Science*, 324(5934), 1519-1520.

5. *Fourth, are the relations modeled the most relevant ones? The authors choose the sectors that “account for more than 90% of the drivers for the majority of the planetary pressures” (line 96). That seems like a good criterion to me. From an economic perspective the absence of a “service sector” seems a strong choice – wouldn’t it in principle be possible that demand and hence factors are re-allocated to such a sector in response to carbon pricing, potentially easing pressures on the environment? I would appreciate if the authors could address the absence of the sector explicitly.*

Reply: We thank the reviewer for raising this very interesting point. It is true that we do not have a service sector and we do agree that this sector is rather sizable in advanced economies. The reason for not explicitly including it in the model is that we wanted to focus on the sectors that are directly relevant to exerting planetary pressures (which the service sector, being mainly labor intensive, is likely not). We also agree that movement of inputs between sectors is potentially important. However, fully capturing such movements would require keeping track of all inputs and the possibility to move them between the sectors. While, for reasons of tractability, we stay away from such a detailed general equilibrium model, we believe that the model we use at least partially captures the aspects you mention in two ways. Firstly, most sectors include “other inputs” that are supplied with an elasticity, meaning that there are adjustments similar to movements of inputs between sectors. In consequence, for the purposes of our analysis (and consistent with the reviewer’s statement above), there is scope for substitution away from “polluting inputs”, potentially reducing the planetary pressures. Secondly, in the manufacturing sector the substitutability between energy and other inputs is parameterised based on the entire economy, which includes the service sector, meaning that our parameter reflects short-run possibilities for substitution present in the economy. Our current set up captures these aspects well enough for our limited purpose. Adding the service sector explicitly would add substantial complexity to the model but would be unlikely to provide much new insight.

We now elaborate on these issues in two places in the revised manuscript. Firstly, on p.17, line 312-316 we now write that “The possibility of adjusting these other inputs leads to decreased use in sectors where their marginal value decreases and increased use in sectors where their marginal value increases, and thus to some extent captures the possibility to move inputs between sectors in response to changing economic conditions.”. Secondly, on p.18, line 336-340 we have added the following text “While we refer to this sector as manufacturing, the substitutability between energy and other inputs is chosen to match that of the economy as a

whole. The substitutability thus reflects not only the manufacturing sector but also the service sector that has a significantly lower energy intensity but that is economically important.”

6. *Overall, I think it would be possible to make reasonable, alternative assumptions that would yield mor even qualitatively different results. For example, one could argue that the carbon price increase foreshadows more stringent future climate policies such that fossil fuel supply does not change or even increases (the green paradox). That is not a limitation of the current paper, but rather a feature of any economic modeling of this kind. For example, the authors refer to the highly influential integrated assessment literature, in which reasonable adjustments in parameters result in the carbon price changing by orders of magnitude. I think the authors here make careful, well-argued choices.*

Reply: Thanks for the compliment! We agree that dynamic aspects are important and including e.g. technical change and uncertainty will indeed yield further insights. As mentioned above, the development of a dynamic multi-sector model capable of this analysis is a key research area for the future.

Specific comments:

- 7. *Title: I would suggest using carbon price rather than tax as it is more general, encompassing cap-and-trade schemes.*

Reply: We agree this is better. We have rephrased the wording in the document, replacing “carbon tax” with “carbon pricing” whenever appropriate. We have also changed the title to “Carbon pricing and Planetary boundaries”.

- 8. *Line 23: The ESPs should be named explicitly from the start, earlier than on page 3.*

Reply: We agree and now list the ESPs starting on p.2., line 22-24

- 9. *Line 24: What is a “catastrophic consequence” in economic terms in this context?*

Reply: This is an excellent question that touches on the main reason for writing this paper. The pressures exerted by the economic system on the earth system risk pushing the earth system from the current equilibrium, around which human societies have been built for the last 10,000 years, into some unknown alternative state. It seems likely that the consequences of this impose intolerably large costs upon society. Hence, the risk of causing such a transition should be a first-order concern for global economic policy. It is in this broad sense that the word “catastrophic” is used in the literature on planetary boundaries. Since our analysis does not rest on any particular quantitative definition of “catastrophic,” we continue using it in this broader conventional sense. We do, however, only use it sparingly and with qualifiers such as “**risking** catastrophic and irreversible global environmental change” and “**potentially** catastrophic consequences.

- 10. Line 26: The “the complexity of the many interlocking processes”: In this paper, no interactions between the ESPs are included?

Reply: The reviewer is correct. No direct interactions between the ESPs are included in our model. We only analyze interactions occurring through market mechanisms. We have clarified this further on p.2, footnote 2.

- 11. Line 49: It might strengthen your argument to explicitly address why you do not want to use a version of the GTAP (or another) general equilibrium model to derive the economic impacts of a carbon price but rather prefer to only use the GTAP data in a new, partial model.

Reply: Our main objective in the analysis was to evaluate the effects of a commonly considered benchmark policy, a carbon tax, within the planetary boundary framework, and to see the degree to which additional policies may be needed to mitigate any potential unintended effects. The planetary boundaries framework dictates much of the scope of our analysis, therefore, and there is currently no integrated global computational model that covers many of the sectors we must consider (e.g. water, fertilizer). Even the GTAP-water model, which is perhaps the closest match, would need to be further extended to help answer our research question. Moreover, because of the GTAP model’s detailed sectoral and regional disaggregation, the planetary boundaries would need to be carefully mapped into the model, a task that would be virtually impossible with presently available information. This is an interesting challenge that we leave for future research.

In addition, our main interest is in gaining some high-level understanding of the key drivers at a global level, consistent with the level of analysis in the PB literature (see refs 1-3 in manuscript). The use of a large-scale computational model, such as the GTAP, would only make this pursuit more difficult, since its rich sectoral and regional (or trade-related) detail, would mainly serve to obscure the mechanisms of most interest to us.

We remark on this point in footnote 5 (p.3) in the revised manuscript.

- Line 50: Please explain what “transparent enough to enable independent assessment of the validity of the results” means in this context. Specifically, how do you define “valid results” here?

Reply: What we meant to state here was that our model is both relatively simple (in terms of being very aggregate in sectoral composition) and transparent, and, by providing the associated Python codes and parameters, readers could easily replicate our results, evaluate their robustness, and determine the main drivers behind them.

In this sense, the word “validity” was meant merely to reflect that it is relatively easy to determine what drives the results and to what extent these drivers are appropriately captured in the model. We have rewritten the text on p.3, lines 41-59 to clarify this and other things (we have removed the word “validity” from the text).

- Line 54: “most ESPs” – please spell out the number exactly.

Reply: We agree that this was unnecessarily vague and specify on (what is now) p.4, lines 60-61 that the policy would “relieve pressure on all ESPs except land and freshwater use”.

- Line 69: “merely 4% of global economic output” undersells the importance of food (the paradox of value).

Reply: We thank the reviewer for reminding us of this paradox. What we wished to convey with that sentence was precisely this: that agriculture, while constituting a small share of GDP, used a disproportionately large share of resources. In the revised manuscript, the sentence is now changed to say: (p.5, line 76-77) “Agricultural activity accounts for 4% of global economic output, uses about 40% of the land surface area, and drives a vast majority of land conversion”.

- Line 82-85: “The design is the conscious result of...” I would argue that is the aim of any analytic economic model and hence the statement maybe is not needed? Or could you be more specific?

Reply: Thanks for pointing this out! We have revised this paragraph (p.5, lines 88-92, current manuscript) to avoid stating the obvious.

- Line 88-91: I think it would be more accurate to write that DICE and IMAGE are models trying to answer different research questions and hence necessarily have a rather different structure.

Reply: Thanks! We agree this is the case. We have now tried to clarify this text which has moved to page 5, footnote 9.

- Line 96: “account for more than 90% of the drivers for the majority of the planetary pressures”. The statement is a little ambiguous. Could you be more specific?

Reply: Thank you for highlighting that this statement was imprecise. For brevity, we chose not to add details at this point in the text but instead added references (p.6, lines 101-102) to table 1 and to the section “Drivers of planetary pressures” in the SI where a detailed description can be found.

- Line 108: In what way are the parameters easier to interpret in this analysis?

Reply: The numerical computations are based on expenditure shares rather than more abstract utility or production function parameters. Hence, the numerical values that go into the actual computations should be much more relatable for most readers and especially those with little prior experience of economic theory. We have now included such a clarification starting on line p.6, line 114-116.

- Line 110: Wouldn't it help the reader if you added a simplified example of how you combine the numerical parameters to get the policy impact for a specific ESP?

Reply: This is what we have tried to do in the numeric results section. Here we go through each planetary boundary and describe how we have derived the change in their underlying drivers based on the change in our model variables as a result of the policy experiments.

- Line 111-112: Again, IAMs answer a different question. Hence the current model cannot be easily compared to them according to these criteria. There are simplified analytic IAMs that allow a similar level of intuitive insight .

Reply: We agree with the reviewer here that our comparison is not quite like-for-like. What we wished to convey here was that we use a linear-approximation-around-the-equilibrium approach, and we merely meant to compare it to what the reader may be more familiar with. On, p.6, line 107-110, we are more explicit about what we do without comparing our approach to the IAMs, which, as you rightly suggest, are a different class of models.

- Line 129: In which sense are the parameter values "coherent" because they stem mostly from the same source?

Reply: Yes, this is what we mean by coherent. They are largely derived from a single data source (GTAP). The remaining are from various sources detailed and referenced in table 5. Since this was unclear we have now replaced the word "coherent" with "internally consistent" which we feel better captures this point (p.8, line 136-138).

- Line 134: You assume counterfactually that the current carbon price in your model is zero, yet there are carbon prices currently in place. How does that affect your interpretation of your results?

Reply: The reviewer raises an interesting point. The first thing to note is that the initial level of the carbon price does not make a significant difference to the qualitative interpretation of our results. In our comparative-statics equations, the price (τ_E) only shows up in equation (S.54). We can see there that the effect of a one percentage point increase of the price has a smaller effect if the initial price is large (since the relative change in $1+\tau_E$ then becomes smaller). Assuming a different initial price would scale all the effects by a common factor, but not fundamentally change their interpretation.

Starting at a different initial carbon price level could potentially affect the results quantitatively, but there are three reasons to think the effect would be small. First, unless we assume that the initial carbon price is very far from zero, the aforementioned scaling factor is close to one. Second, we note that all other parameter values that we use are empirically derived and hence reflect their values under the currently prevailing carbon price. Finally, once you subtract the

\$270 billion spent annually on fossil fuel subsidies (based on latest IMF figures) from globally weighted average price per tonne of carbon (based on this world bank report), you end up with a net global carbon price that is quite close to zero, and probably slightly negative. We believe a zero global carbon price is a reasonable first-order approximation.

We do, however, certainly agree that the lack of a description of this could cause confusion and have therefore added a discussion of this starting on p.22, line 397-403 and added footnote 10 on page 8 that points the reader to this discussion.

- Line 142: It would be nice if you could compare your elasticity of CO2 emissions with respect to the carbon price to other existing estimates for specific sectors or countries to see if your model produces results that are similar to empirical estimates

Reply: This is a most relevant comparison to make. We have looked at other models and empirical analyses of the effects of carbon prices on emissions. A detailed description of this can be found in the new section “Elasticity of emissions to the carbon price” in the SI. In the main manuscript we added a brief description of and reference to these results on p.9, line 151-154. The conclusion is that emissions are less sensitive to the tax than in some other models, but that it seems to be relatively well-aligned with empirical estimates.

- Figure 2: Please explain the colors in the figure.

Reply: The colors are the same as in the original “Planetary boundaries” papers. We have added a description to the captions of figures 2 and 3 in the revised manuscript.

- Line 162: “...a combination of a number of different effects”. Under “Drivers of planetary pressures”, the SI only mentions the same ones as the text? The SI does not feature a section “Determining how policy-induced changes impact the planetary pressures”.

Reply: Thank you for pointing this out! This section is actually under “Numerical results” in the “Methods section” (and not in the SI). We have now fixed this reference (p.9, lines 172-173).

- Line 176: “...increased freshwater use...” I consider the freshwater sector a good case for which the authors could explain the impact of their simplifying assumptions intuitively. For example, not all farms around the world have access to irrigation and hence cannot substitute fossil fuel for freshwater .

Reply: We thank the reviewer for raising an interesting point that was not clear from our brief description. The increased freshwater use results from a slight increase in the relative price of almost all other agricultural inputs (not just energy input), consequent to a carbon tax. In consequence, even if not all farms have irrigation, and hence cannot substitute fossil fuel for freshwater, they end up using slightly more water (in our model) to substitute for using less of

other inputs e.g. fertilizers.¹ Clearly, there is a regional component here, with many regions being unable to increase water use to compensate for any relative price increases of substitute inputs, in which case output adjustments must occur. This is similar to the considerations we dealt with in our answer to Q(4) above.

We have added a brief explanation that intuitively explains some of these aspects in the revised manuscript, p. 11, line 188-194, stating “ The effect of the carbon tax upon freshwater use illustrates the implications of global aggregation in modeling: our global aggregate model responds with increased aggregate water use. In reality, with not all farms across the world being able to substitute freshwater for energy-intensive inputs (e.g. in subsistence farming with no irrigation), either other inputs or output must adjust. More explicitly, a carbon tax in this scenario may leave freshwater use unchanged but lead to either increases in carbon emissions or other planetary pressures (if other inputs are substituted) or a reduction in output.”.

- Line 178-9: I think it would be good to point out the relative magnitude of the effects for all ESPs that are not directly impacted but via linkages between economic sectors.

Reply: This is an important point. We have now added a paragraph starting on p.11, line 195-204 discussing this.

- Line 204-5: “Other types of biofuels, that are produced differently, would perhaps fit better into our renewables-category”. Could you state an estimate of the share of biofuels currently on the market that are competing with food for land? Also, it might be worth pointing out that for example the EU Renewable Energy Directive states the type of land that may be converted to biofuel production , potentially already circumventing the problem identified here?

Reply: We thank the reviewer for raising this interesting point and suggesting a clarification based upon the increasing use of sustainability criteria in biofuels policies. The reviewer’s first point, the share of biofuels currently on market competing with food for land, can be estimated approximately based on the literature: estimates for 2013 suggest that 2-3% of global cropland (between 32-48 Million hectares, of a total of 1600 Million hectares) is used for biofuel. Other estimates (for 2015) suggest a range of between 15 and 41 Mha (see Goetz et al. (2018), §3.5). Overall, the estimate of the share of biofuels competing with food for land lies in the range of 1-3%.

As to the second point, as the reviewer suggests, the recast Renewable Directive (2018/2001), already specifies criteria limiting the counting of biofuels derived from food and feed crops² towards any EU member country’s renewable energy goals. Adoption of these strategies

¹ We note that the other non-modelled input in agriculture, denote M_A, is not very substitutable (and neither are other non-land inputs, with an elasticity of 0.75, see Table 5), meaning that it cannot completely absorb the effect of changes in energy intensive inputs, and some increase in fresh water use results as a consequence.

² [On a sliding scale, with a reduction in, starting 2023, to zero by 2030. See https://ec.europa.eu/commission/presscorner/detail/en/MEMO_19_1656](https://ec.europa.eu/commission/presscorner/detail/en/MEMO_19_1656)

worldwide (for which there is not much evidence, admittedly) should eventually limit the problem identified here.

In consequence, we have modified the text in the relevant lines (p.14, lines 228-234, revised manuscript) to read instead: “This analysis assumes biofuels produced using the same inputs as food and feed, reflecting current production patterns (with biofuel production using between 1-3% of total cropland area, see Goetz et al. (2018), §3.5). When biofuels of this type are phased out in favour of those not competing directly with food crops for land, either by policy (as required by a new EU Renewable Directive) or technology change (so-called second generation biofuels), then biofuels can be folded into our renewable energy category.”

- *Line 220: The table reference is broken (as are others in the SI).*

Reply: We have now fixed this.

- *Line 243-244 “Our analysis clearly shows that indeed these interacting effects can be quite substantial and should not be ignored”. Where the effects are not “direct” (as they are for ocean acidification and climate change, -0.25%), the effects are at least one order of magnitude smaller (biodiversity being the largest with 0.027%). The exception are nitrogen flows (-0.14%, where link is “rather direct” as natural gas is its most important input by far).*

Reply: We have now changed it to “We can see that while the indirect effects are significantly smaller than the direct effects, they are far from insignificant.” (p.15, line 268-270). See also our response to the question raised above for (regarding *Line 178-9*).

- *Line 363: “About 50”. Please state the exact number.*

Reply: This was a preliminary phrase that should have been replaced before submission. Thanks for catching this. The true number is 39. We have now fixed this in the document.

REVIEWERS' COMMENTS:

Reviewer #1 (Remarks to the Author):

Thanks. I am convinced with the corrections and the explanations in the revised version.

Reviewer #2 (Remarks to the Author):

I thank the authors for their care- and thoughtful replies to my admittedly extensive remarks. I hope they (eventually) consider the process to have improved the manuscript. I am fully satisfied with their replies and with the adjustments they have made in the manuscript.

I believe the manuscript addresses an important economic question, it is well-written, and it pleasingly does not claim to do more than what it actually does (which is a lot). I hope that it inspires other economists to think about its content in the future and to write responses that expand and challenge its results.

I will refrain from replies to the replies as they would at this point result in second order effects. It is after all not me writing the paper. In the same spirit, I hope the editor together with the authors can re-assess the readability of the paper at this point. It would be a shame if addressing the reviewers' every little pet peeve would come at the cost of distracting the readers from the main message.